# Bayesian Weight Enhancement with Steady-State Adaptation for Test-time Adaptation in Dynamic Environments

**Jae-Hong Lee** [1]

## Abstract

Test-time adaptation (TTA) addresses the machine learning challenge of adapting models to unlabeled test data from shifting distributions in dynamic environments. A key issue in this online setting arises from using unsupervised learning techniques, which introduce explicit gradient noise that degrades model weights. To invest in weight degradation, we propose a Bayesian weight enhancement framework, which generalizes existing weight-based TTA methods that effectively mitigate the issue. Our framework enables robust adaptation to distribution shifts by accounting for diverse weights by modeling weight distributions. Building on our framework, we identify a key limitation in existing methods: their neglect of time-varying covariance reflects the influence of the gradient noise. To address this gap, we propose a novel steady-state adaptation (SSA) algorithm that balances covariance dynamics during adaptation. SSA is derived through the solution of a stochastic differential equation for the TTA process and online inference. The resulting algorithm incorporates a covariance-aware learning rate adjustment mechanism. Through extensive experiments, we demonstrate that SSA consistently improves state-of-the-art methods in various TTA scenarios, datasets, and model architectures, establishing its effectiveness in instability and adaptability.

## 1. Introduction

Machine learning algorithms have achieved remarkable success due to the ability of deep neural networks (DNNs) to model large-scale data (Krizhevsky et al., 2012; Simonyan & Zisserman, 2014; He et al., 2016; Hinton et al., 2012). This success rests on the critical assumption that the test data adhere to the same distribution as the train data used to pre-train the source model (Goodfellow et al., 2016; Murphy, 2023). However, in real-world applications, models often encounter dynamic environments in which the data distribution changes over time, making this assumption difficult to hold (Hendrycks & Dietterich, 2019b; Koh et al., 2021). For example, autonomous driving systems face external factors such as weather variability and internal issues such as sensor degradation, resulting in substantial performance degradation. Even models with robust training pipelines often exhibit high sensitivity to distribution shifts, with minor deviations causing a significant drop in accuracy (Quinonero-Candela et al., 2008; Sun et al., 2017). Consequently, robust online adaptation mechanisms are essential, ensuring reliable performance in dynamic settings.

Test-time adaptation (TTA) is an online adaptation technique that updates a model using unlabeled test samples drawn from a new data distribution through unsupervised learning. TTA methods often rely on the model's own predictions to drive adaptation, employing entropy minimization loss (Wang et al., 2020). However, in dynamic environments with complex, time-varying distribution shifts, these unsupervised approaches face significant challenges. One critical issue is weight degradation, where reliance on explicit gradient noise—caused by inaccurate model predictions—corrupts the weights over time (Boudiaf et al., 2022; Chen et al., 2022; Gong et al., 2022; Niu et al., 2023). Recent research has introduced weight-based TTA methods that mitigate this issue by enhancing model weights during adaptation (Wang et al., 2022; Niu et al., 2022; 2023; Marsden et al., 2023; Lee & Chang, 2024), typically by averaging current model weights with the pre-trained source weights.

In this study, we propose a Bayesian weight enhancement framework that unifies and generalizes existing weight-based TTA methods by explicitly modeling the enhanced weight distribution. Our framework, rooted in Bayesian deep learning (Murphy, 2023), offers a principled approach to improving robustness to distribution shifts. Under this framework, existing weight-based methods can be viewed as special cases that assume time-invariant covariance in

[1]Division of Language & AI, Hankuk University of Foreign Studies, Seoul, Republic of Korea. Correspondence to: Jae-Hong Lee <ljh93ljh@hufs.ac.kr>.

*Proceedings of the 42nd International Conference on Machine Learning*, Vancouver, Canada. PMLR 267, 2025. Copyright 2025 by the author(s).

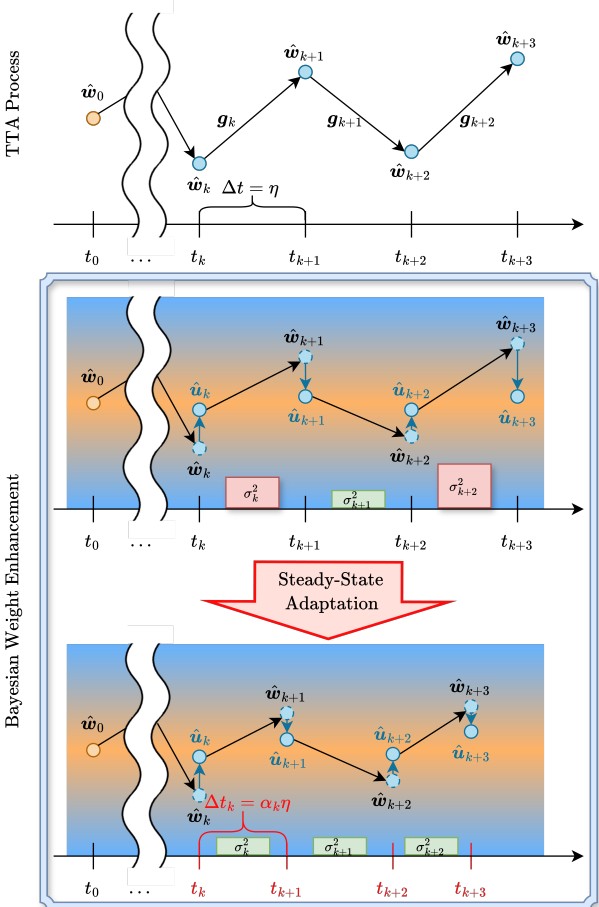

*Figure 1.* (**Top**) Test-time adaptation process. The blue circles indicate that the weight, $\hat{\boldsymbol{w}}_k$, evolves over time. The black arrows denote gradients, $g_k$, emitted at arbitrary discrete time steps $t_k \in \{t_0, t_1, \ldots, t_K\}$. (**Bottom**) Bayesian weight enhancement with steady-state adaptation. Bayesian weight enhancement aligns weights to the source model $\hat{\boldsymbol{w}}_0$ for enhanced weight $\hat{\boldsymbol{u}}_k$. Steady-state adaptation balances the variance $\sigma_k^2$ and adjusts the step size $\Delta t_k$ by calculating $\alpha_k$ from the variance.

the weight distribution. However, the TTA methods in dynamic environments introduce the explicit noise arising from changes in the data distribution, and covariance reflects the evolution of the noise over time. This covariance represents the stability and adaptability of the learning process: excessive covariance destabilizes learning by overshooting optimal solutions, while insufficient covariance reduces adaptability, trapping the model in suboptimal states (Jastrzebski et al., 2017; Zhu et al., 2018; Wu et al., 2020). These challenges make balancing covariance a critical aspect of improving TTA performance.

To address these challenges, we draw on stochastic differential equation (SDE) approximations (Li et al., 2019; 2021), which provide a theoretical foundation for modeling

the dynamics of weight distributions in stochastic gradient descent (SGD). Using this foundation, we propose the steady-state adaptation (SSA) algorithm, which dynamically regulates covariance during TTA to ensure stability and adaptability. SSA is derived in three steps: (1) modeling the transition weight distribution via the SDE approximation of SGD, (2) performing online inference for the posterior weight distribution using this transition model, and (3) adjusting weight updates to maintain steady-state covariance. Our algorithm introduces a covariance-aware learning rate adjustment mechanism that increases learning rates under low noise for enhanced adaptability and decreases them under high noise for improved stability. We validate SSA in various TTA scenarios, demonstrating its consistent improvements over state-of-the-art methods. In addition, SSA addresses common challenges such as performance degradation in increasing learning rates (Zhao et al., 2023), providing a robust and principled solution in dynamic environments.

## 2. Background

In this section, we summarize previous studies as the basis for our proposed framework and algorithm. Related works are provided in Appendix 6.

**Weight-Based Test-time Adaptation** TTA is an online learning paradigm designed to adapt a pre-trained source model to unlabeled test samples drawn from a shifted distribution. However, traditional TTA methods are prone to weight degradation in dynamic environments with evolving diverse distribution shifts (Boudiaf et al., 2022; Gong et al., 2022; Niu et al., 2022; Lee et al., 2024). This degradation occurs because explicit gradient noise from unsupervised learning accumulates damage to the model weights over time. Weight-based TTA methods provide a complementary approach by directly addressing weight degradation. Instead of relying solely on noisy gradient updates, these methods continually integrate the pre-trained source weights into the current model during adaptation. At each time step $k$, the current weights $\hat{\boldsymbol{w}}_k$ are adjusted by averaging them with the pre-trained source weights $\hat{\boldsymbol{w}}_0$ as follows:

$$\hat{\boldsymbol{w}} = a\hat{\boldsymbol{w}}_k + (1-a)\hat{\boldsymbol{w}}_0, \tag{1}$$

where $0 \leq a \leq 1$ is a weighting factor that determines the balance between the current and source weights. Several state-of-the-art weight-based TTA methods adopt this principle: SAR (Niu et al., 2023) resets the model weights to the source weights (i.e., $a = 0$) when the exponential moving average of the total loss falls below a predefined threshold. ROID (Marsden et al., 2023) employs a moving average approach with a small $a$, anchoring the weights close to the source weights. CMF (Lee & Chang, 2024) introduces a "hidden source weight" by averaging the source and model

weights over time with an additional parameter similar to $a$. This hidden weight is then further averaged with the current model weights to produce $\hat{w}$. These methods improve TTA performance in dynamic environments by mitigating weight degradation during adaptation.

**Approximating SGD with SDEs** SGD is a cornerstone of deep learning optimization, and its behavior can be effectively analyzed by approximating its discrete updates using SDEs (Li et al., 2019; 2021; Malladi et al., 2022). SDE approximations offer a theoretical framework for how the gradient noise of covariance influences optimization dynamics, convergence, and generalization. The introduction of SDE approximations provided key insights into the average behavior of SGD, linking covariance to convergence properties (Li et al., 2019). Subsequent work validated these approximations for discrete SGD updates, showing how learning rate adjustments influence covariance and, in turn, the optimization trajectory (Malladi et al., 2022). Specifically, increasing learning rates amplify the noise, promoting exploration of flatter minima, while lower learning rates suppress the noise, stabilizing optimization but risking convergence to sharp minima. These studies are particularly relevant for TTA, where shifting data distributions cause excessive changes in covariance, significantly affecting the performance of the model. SDE approximations thus serve as a powerful tool for understanding and addressing these covariance dynamics, providing a foundation for robust optimization in the presence of distribution shifts.

## 3. Methodology

In this section, we start by introducing the notation and problem formulation, followed by the Bayesian weight enhancement framework and the proposed SSA algorithm. The framework with our algorithm is presented in Algorithm 1.

### 3.1. Problem Formulation

We consider a DNN model $f : \mathcal{X} \to \mathcal{Y}$, parameterized by an arbitrary weight $\hat{w} \in \mathbb{R}^d$, which maps input data $x \in \mathcal{X}$ to a predicted label probability $p(y|x, w)$ where $y \in \mathcal{Y}$ and $d$ is the dimensionality of the weight space. Given a well-trained source model $f(.; \hat{w}_0)$ pre-trained on labeled source data $\{(x_n, y_n) \sim D_0 : n = 1 : N_0\}$, the objective of TTA is to adapt the model to unlabeled test data $x_k \sim D_k$, where $D_k$ represents a dynamically evolving new distribution. This adaptation is required at each discrete time step $k \in \{1, 2, 3, \ldots, K\}$, under $D_k \neq D_0$.

In the TTA process, the unsupervised objective is the expected risk $G(\cdot)$ at each time step $k$, defined as:

$$G(\hat{w}_k) = \mathbb{E}_{x_{k+1} \sim D_{k+1}}\big[\ell\big(f(x_{k+1}; \hat{w}_k)\big)\big], \quad (2)$$

---

**Algorithm 1** Bayesian Weight Enhancement Framework with Steady-State Adaptation

---

**Require:** Learning rate $\eta$, Source model $f(.; \hat{w}_0)$, $A = a\mathbf{I}$, $\sigma_\lambda^2$

Initialization $\mathbf{m}_0 = \hat{w}_0, \mathbf{P}_0 = \mathbf{0}, \hat{g}_0 = 0$

**for** $k = 1$ **to** $K$ **do**

  $g_k \leftarrow \nabla_{m_k} G(m_k)$

  **Steady-state adaptation:**

  $\bar{g}_k \leftarrow \frac{1}{k}(g_k + \hat{g}_{k-1})$

  $\sigma_k^2 \leftarrow \frac{1}{d}\text{tr}\Big((g_k - \bar{g}_k)(g_k - \bar{g}_k)^\top\Big)$

  $\alpha_k \leftarrow \sqrt{\sigma_\lambda^2/(\eta^2 \sigma_k^2)}$        ▷ Eq. (19)

  $m_{k+1|k}^+ \leftarrow m_k - \alpha_k \eta g_k$       ▷ Eq. (20)

  $\mathbf{P}_{k+1|k}^+ \leftarrow \mathbf{P}_k + \alpha_k^2 \eta^2 \Sigma_k$      ▷ Eq. (21)

  $\hat{g}_k \leftarrow (\alpha_k g_k + \hat{g}_{k-1})$

  **Bayesian weight enhancement:**

  $m_{k+1} \leftarrow m_{k+1|k}^+ + A(\hat{w}_0 - m_{k+1|k}^+)$   ▷ Eq. (17)

  $\mathbf{P}_{k+1} \leftarrow (\mathbf{I} - A)\mathbf{P}_{k+1|k}^+$       ▷ Eq. (18)

**end for**

---

where $\ell(.)$ is typically an entropy-based loss function:

$$\ell\big(f(x_{k+1}; \hat{w}_k)\big) = -\sum_{y \in \mathcal{Y}} p(y|x_{k+1}, w_k) \log p(y|x_{k+1}, w_k).$$
$$(3)$$

The predictive distribution $p(y|x_k, w_k)$ represents the model output for input $x_k$, where the weights are iteratively optimized via SGD:

$$\hat{w}_{k+1} = \underset{\hat{w}_k}{\arg\min}\, G(\hat{w}_k). \quad (4)$$

This sequential weight update process produces a time series of weights $\{\hat{w}_1, \hat{w}_2, \ldots, \hat{w}_k\}$. However, due to the absence of ground-truth labels in the TTA settings, the reliance on unsupervised learning introduces explicit noise into the gradient estimates. This explicit gradient noise accumulates over time, leading to weight degradation, which ultimately compromises model performance, especially in dynamic environments characterized by significant and evolving distribution shifts.

The primary goal of this paper is to mitigate the effects of the explicit noise caused by the unsupervised nature of the TTA process. To achieve this, we aim to derive an enhanced weight sequence $\{\hat{u}_1, \hat{u}_2, \ldots, \hat{u}_k\}$ where $\hat{u}_k$ represents a refined weight estimate at the time step $k$. This enhanced weight sequence is designed to restore and improve model performance under dynamic distribution shifts, offering a robust solution to the challenges of weight degradation.

## 3.2. Bayesian Weight Enhancement with Time-Invariant Covariance

Bayesian deep learning provides a principled approach to represent model weights as probability distributions rather than deterministic values (Polson & Sokolov, 2017; Wilson, 2020; Wilson & Izmailov, 2020; Khan, 2019). This representation enables the consideration of multiple plausible weight configurations simultaneously, which enhances the robustness of distribution shifts and the explicit gradient noise (Wortsman et al., 2022; Rame et al., 2022). These properties make Bayesian approach particularly well suited for the TTA methods, where models adapt to evolving data distributions in dynamic environments while maintaining stability against noise.

Using Bayesian model averaging, the posterior predictive distribution is expressed as:

$$p(y|\boldsymbol{x}_k, \boldsymbol{w}_k, \boldsymbol{w}_{k+1}) = \int p(y|\boldsymbol{x}_k, \boldsymbol{u})p(\boldsymbol{u}|\boldsymbol{w}_k, \boldsymbol{w}_{k+1})d\boldsymbol{u}, \quad (5)$$

where $p(\boldsymbol{u}|\boldsymbol{w}_k, \boldsymbol{w}_{k+1}) \propto p(\boldsymbol{w}_{k+1}|\boldsymbol{u})p(\boldsymbol{u}|\boldsymbol{w}_k)$. To compute the posterior weight distribution $p(\boldsymbol{u}|\boldsymbol{w}_k, \boldsymbol{w}_{k+1})$, we approximate both likelihood and prior terms using linear Gaussian models:

$$p(\boldsymbol{w}_{k+1}|\boldsymbol{u}) = \mathcal{N}(\boldsymbol{w}_{k+1}|\hat{\boldsymbol{w}}_0, R), p(\boldsymbol{u}|\boldsymbol{w}_k) = \mathcal{N}(\boldsymbol{u}|\hat{\boldsymbol{w}}_k, Q), \quad (6)$$

where $R = \sigma_r^2\mathbf{I}$, $Q = \sigma_q^2\mathbf{I}$ and $\mathbf{I} \in \mathbb{R}^{d \times d}$ is the identity matrix. The formulation results in the posterior distribution that is also Gaussian:

$$p(\boldsymbol{u}|\boldsymbol{w}_k, \boldsymbol{w}_{k+1}) = \mathcal{N}(\boldsymbol{u}_k|\boldsymbol{m}, \mathbf{P}). \quad (7)$$

The posterior mean and covariance are given by:

$$\boldsymbol{m} = (\mathbf{I} - A)\hat{\boldsymbol{w}}_k + A\hat{\boldsymbol{w}}_0, \ \mathbf{P} = AR, \quad (8)$$

and $A = Q(Q + R)^{-1}$. Recent weight-based TTA methods, such as ROID and CMF, have achieved performance improvements by setting $A$ close to $\mathbf{0}$. This setting corresponds to the case where $\sigma_q^2 \ll \sigma_r^2$. Furthermore, for a well-trained source model with $\sigma_r^2 \to 0$, the covariance $\mathbf{P}$ becomes $\mathbf{0}$. In the limit where $\mathbf{P} \to \mathbf{0}$, the posterior distribution $p(\boldsymbol{u}_k|\boldsymbol{w}_0, \boldsymbol{w}_k)$ reduces to a Dirac delta $\delta(\boldsymbol{u}_k - \boldsymbol{m})$. Under these conditions, the posterior predictive distribution in Eq. (5) is simplified to $p(y|\boldsymbol{x}_k, \boldsymbol{m})$, where $\boldsymbol{m}$ represents the updated weight that combines information from both the current and the source weights.

From the perspective of our framework, existing weight-based TTA methods can be interpreted as specific cases of the posterior weight distribution under the assumption of time-invariant covariance. However, dynamic environments frequently induce variability in model performance due to data distribution shifts, which result in changes in gradient

noise over time. This evolving noise profile directly affects the covariance of the weight distribution (Jastrzebski et al., 2017; Zhu et al., 2018; Wu et al., 2020), capturing the trade-off between adaptability and stability during training.

## 3.3. Steady-State Adaptation

In this section, we introduce the SSA algorithm, which integrates the dynamics of SGD using an SDE approximation, applies Bayesian filtering to compute the posterior weight distribution, and ensures steady-state covariance. The detailed derivations are provided in Appendix A.

**Transition Weight Distribution with Time-varying Covariance** In many TTA methods, the SGD optimizer serves as the backbone to update model weights. We adopt the SDE approximation to capture the temporal dynamics of the weight and their covariance during the TTA process (Li et al., 2019). This approximation provides a continuous-time stochastic representation of SGD. For a small learning rate $\eta$, the weight dynamics during TTA can be expressed as:

$$d\boldsymbol{u}_t = -g_t dt + \sqrt{\eta}\Sigma_t^{1/2}dW_t, \quad (9)$$

where $dW_t$ is standard Brownian motion, $g_t$ represents the gradient of the loss function $\nabla G(\hat{\boldsymbol{u}}_t)$, and $\Sigma_t = \sigma_t^2\mathbf{I}$ describes the covariance matrix derived from the gradient (Malladi et al., 2022). The time-varying variance $\sigma_t^2$ is computed as $\sigma_t^2 = \frac{1}{d}\text{tr}\left(\frac{1}{t}\sum_{\tau=1}^{t}(g_\tau - \bar{g}_\tau)(g_\tau - \bar{g}_\tau)^\top\right)$, where $\bar{g}_t = 1/t\sum_{\tau=1}^{t}g_\tau$ is the mean gradient, $\text{tr}(\cdot)$ is the trace operator. Covariance reflects the variability of the gradient and serves as an indicator of optimization stability.

The temporal evolution of the weight distribution $p(\boldsymbol{u}_t)$ is governed by the Fokker-Planck-Kolmogorov (FPK) equation:

$$\frac{\partial p(\boldsymbol{u}_t)}{\partial t} = \sum_{i=1}^{d}\frac{\partial p(\boldsymbol{u}_t)}{\partial w_t^i}[g_t]_i + \frac{1}{2}\sum_{i=1}^{d}\sum_{j=1}^{d}\frac{\partial^2 p(\boldsymbol{u}_t)}{\partial w_t^i \partial w_t^j}\eta[\Sigma_t]_{ij}, \quad (10)$$

where $[\cdot]_i$ and $[\cdot]_{ij}$ represent the $i$-th element of a vector $[\cdot]$, and $(i, j)$-th element of a matrix $[\cdot]$, respectively. Since $g_t$ and $\Sigma_t$ are generally intractable, we solve the FPK equation by assuming a Gaussian approximation $p(\boldsymbol{u}_t) \approx \mathcal{N}(\boldsymbol{u}_t|\boldsymbol{m}_t, \mathbf{P}_t)$ and applying a Taylor expansion to linearize the dynamics. As the results, the mean and covariance evolve as follows:

$$\frac{d\boldsymbol{m}}{dt} = -g_t, \ \frac{d\mathbf{P}}{dt} = \mathbf{P}\mathcal{G}_t^\top + \mathcal{G}_t\mathbf{P} + \eta\Sigma_t, \quad (11)$$

and $\mathcal{G}_t$ is the Jacobian matrix of $g_t$ w.r.t. $\boldsymbol{u}_t$. In this situation, the transition distribution $p(\boldsymbol{u}_t|\boldsymbol{u}_s)$ is derived by solving this equation for the interval $0 < s < t$ and the initial conditions $\boldsymbol{m}_s = \hat{\boldsymbol{w}}_k$ and $\mathbf{P}_s = \mathbf{0}$ (Särkkä & Solin, 2019). For a small

interval $s = k\eta$ and $t = (k+1)\eta$, we can approximate the gradients and covariance as a constant within each step $k$, leading to $\mathcal{G}_t = 0$. In this situation, the transition weight distribution is derived as:

$$p(\boldsymbol{u}_{k+1}|\boldsymbol{u}_k) \approx \mathcal{N}(\boldsymbol{u}_{k+1}|\boldsymbol{m}_{k+1|k}, \mathbf{P}_{k+1|k}), \quad (12)$$

where $\boldsymbol{m}_{k+1|k} = \hat{\boldsymbol{w}}_k - g_k\Delta t, \mathbf{P}_{k+1|k} = \sigma_k^2\Delta t^2\mathbf{I}$ with $g_k = \nabla G(\hat{\boldsymbol{w}}_k)$, $\sigma_k^2 = \frac{1}{d}\mathrm{tr}\left((g_k - \bar{g}_k)(g_\tau - \bar{g}_k)^\top\right)$, and the step size is $\Delta t = \eta$.

This transition distribution aligns the evolution of the mean with the discrete TTA process, while the covariance reflects the influence of gradient noise. The covariance serves as a measure of noise shape in the weight updates, highlighting the interplay between noise and optimization dynamics during TTA.

**Online Posterior Weight Distribution Inference**
To perform an online inference of the posterior weight distribution in our Bayesian weight enhancement framework, we adopt a Bayesian filtering approach (Särkkä & Svensson, 2023). The approach provides a recursive mechanism for computing the posterior distribution at each discrete time step by incorporating the prior distribution and the likelihood of new observations. This mechanism enables efficient computation of the posterior distributions in online learning.

Given the posterior weight distribution at the previous time step, $p(\boldsymbol{u}_k|\boldsymbol{w}_{0:k})$, the one-step-ahead posterior distribution $p(\boldsymbol{u}_{k+1}|\boldsymbol{w}_{0:k})$ is computed using the Chapman-Kolmogorov equation (Murphy, 2023):

$$\begin{aligned} p(\boldsymbol{u}_{k+1}|\boldsymbol{w}_{0:k}) &= \int p(\boldsymbol{u}_{k+1}|\boldsymbol{u}_k)p(\boldsymbol{u}_k|\boldsymbol{w}_{0:k})\mathrm{d}\boldsymbol{u}_k \\ &= \mathcal{N}(\boldsymbol{u}_{k+1}|\boldsymbol{m}_{k+1|k}^+, \mathbf{P}_{k+1|k}^+), \end{aligned} \quad (13)$$

where $p(\boldsymbol{u}_{k+1}|\boldsymbol{u}_k)$ is the transition distribution given in Eq. (12). Starting from the initial condition $\boldsymbol{m}_0 = \hat{\boldsymbol{w}}_0$ and $\mathbf{P}_0 = 0$, the mean and covariance are updated as:

$$\boldsymbol{m}_{k+1|k}^+ = \boldsymbol{m}_k - g_k\Delta t, \quad (14)$$

$$\mathbf{P}_{k+1|k}^+ = \mathbf{P}_k + \Sigma_k\Delta t^2. \quad (15)$$

Using our Bayesian framework, the likelihood $p(\boldsymbol{w}_{k+1}|\boldsymbol{u}_{k+1})$ is modeled as $p(\boldsymbol{w}_{k+1}|\boldsymbol{u})$ in Eq. (6), the posterior weight distribution is derived as:

$$\begin{aligned} p(\boldsymbol{u}_{k+1}|\boldsymbol{w}_{0:k+1}) &= \frac{1}{Z_k}p(\boldsymbol{w}_{k+1}|\boldsymbol{u}_{k+1})p(\boldsymbol{u}_{k+1}|\boldsymbol{w}_{0:k}) \\ &= \mathcal{N}(\boldsymbol{u}_{k+1}|\boldsymbol{m}_{k+1}, \mathbf{P}_{k+1}), \end{aligned}$$
$$(16)$$

where $Z_k = \int p(\boldsymbol{w}_{k+1}|\boldsymbol{u}_{k+1})p(\boldsymbol{u}_{k+1}|\boldsymbol{w}_{0:k})\mathrm{d}\boldsymbol{u}_{k+1}$. The mean and covariance of this distribution are updated as:

$$\boldsymbol{m}_{k+1} = \boldsymbol{m}_{k+1|k}^+ + A_k(\hat{\boldsymbol{w}}_0 - \boldsymbol{m}_{k+1|k}^+), \quad (17)$$

$$\mathbf{P}_{k+1} = (\mathbf{I} - A_k)\mathbf{P}_{k+1|k}^+, \quad (18)$$

where $A_k$ is the Kalman gain (Särkkä & Svensson, 2023) defined as $A_k = \mathbf{P}_{k+1|k}^+(\mathbf{P}_{k+1|k}^+ + R)^{-1}$. The term $A_k$ adjusts the influence of the source weight $\hat{\boldsymbol{w}}_0$ based on the relative magnitude of the covariance $\mathbf{P}_{k+1|k}^+$ and $R$. This adjustment ensures that the posterior mean incorporates both the current weight and the source weight, similar to the exist weight-based TTA.

In the TTA process, the gradients $g_k$ are estimated from unlabeled test samples, leading to explicitly noisy covariance estimates $\Sigma_k$. This explicit noise introduces fluctuations in the one-step-ahead covariance $\mathbf{P}_{k+1|k}^+$ and, consequently, $A_k$. Such fluctuations destabilize the weight update process (Kloeden et al., 1992), impairing the ability to adapt effectively to new data samples, particularly under significant distribution shifts. These challenges underscore the need for a robust mechanism to regulate covariance dynamics.

**Balancing Covariance** TTA methods are sensitive to learning rate adjustments. High learning rates exacerbate covariance, leading to abrupt performance drops due to instability (Zhao et al., 2023). In contrast, excessively low learning rates can trap the model in local optima, resulting in significant performance degradation (Niu et al., 2023; Lee et al., 2024). To address these competing challenges, we propose a dynamic algorithm that calculates a step size $\Delta t_k$ to balance covariance based on the posterior weight distribution.

To achieve this balance, we introduce a scalar $\alpha_k > 0$ to parameterize the step size as $\alpha_k\eta$. The dynamic step size is computed to satisfy the condition $\mathbf{P}_{k+1} \approx \mathbf{P}_k$, ensuring that the covariance remains near its steady-state. Given that all covariance matrices in Eqs. (15) and (18) are scalar multiples of the identity matrix, i.e., $\mathbf{P}_k = \mathrm{p}_k\mathbf{I}$, $\Sigma_k = \sigma_k^2\mathbf{I}$ and $R = \sigma_r^2\mathbf{I}$, the steady-state condition is simplified to:

$$\alpha_k = \sqrt{\frac{\sigma_\lambda^2}{\eta^2\sigma_k^2}}, \quad (19)$$

where $\sigma_\lambda = \sqrt{\mathrm{p}_\infty/(\sigma_r^2 - \mathrm{p}_\infty)}$, and $\mathrm{p}_\infty$ represents $\mathrm{p}_k$ in the steady-state regime. Substituting $\alpha_k\eta$ into $\Delta t$ in Eqs. (14) and (15), the enhanced mean and covariance of the one-step-ahead posterior distribution are derived as follows:

$$\boldsymbol{m}_{k+1|k}^+ = \boldsymbol{m}_k - \alpha_k\eta g_k, \quad (20)$$

$$\mathbf{P}_{k+1|k}^+ = \mathbf{P}_k + \alpha_k^2\eta^2\Sigma_k. \quad (21)$$

These updates are then applied to Eqs. (17) and (18) to calculate the posterior weight distribution. Under the steady-state condition, $A_k$ in Eq. (17) stabilizes and becomes $A = a\mathbf{I}$

*Table 1.* Average error rates (%) and standard deviations in covariate shift ($\gamma = \infty$) on ImageNet-C. Red fonts indicate performance degradation with respect to Source.

| Method | NOISE | | | BLUR | | | | WEATHER | | | | DiGiTAL | | | | Avg. |
|---|---|---|---|---|---|---|---|---|---|---|---|---|---|---|---|---|
| | gaussian | shot | impulse | defocus | glass | motion | zoom | snow | frost | fog | bright | contrast | elastic | pixelate | jpeg | |
| Source | 43.9 | 43.3 | 43.4 | 69.7 | 78.3 | 59.6 | 69.1 | 40.1 | 44.3 | 36.3 | 26.5 | 50.6 | 67.6 | 60.6 | 43.4 | 51.8 |
| TENT | 43.8 | 42.9 | 43.1 | 70.1 | 77.9 | 59.4 | 69.3 | 42.2 | 48.7 | 45.9 | 28.9 | 50.4 | 68.0 | 63.3 | 45.3 | 53.3±0.22 |
| LAME | 45.4 | 43.7 | 44.5 | 72.1 | 90.7 | 60.6 | 89.3 | 91.5 | 96.5 | 99.7 | 26.7 | 95.9 | 96.1 | 63.3 | 44.7 | 70.7±0.14 |
| RoTTA | 43.9 | 43.3 | 43.3 | 69.7 | 77.8 | 59.4 | 68.7 | 39.8 | 42.5 | 35.9 | 26.2 | 49.8 | 66.6 | 60.2 | 43.4 | 51.4±0.02 |
| SAR | 44.2 | 43.8 | 43.7 | 69.7 | 77.5 | 57.1 | 66.8 | 41.2 | 41.4 | 42.0 | 26.4 | 51.6 | 64.3 | 57.2 | 42.1 | 51.3±0.35 |
| EATA | 44.3 | 43.8 | 43.5 | 69.0 | 74.3 | 56.8 | 64.2 | 39.8 | 44.2 | 46.3 | 25.4 | 45.3 | 61.2 | 54.1 | 40.7 | 50.2±0.06 |
| DeYO | 43.6 | 41.9 | 40.8 | 66.3 | 70.7 | 54.7 | 63.4 | 38.3 | 38.3 | 36.8 | 24.5 | 43.2 | 55.5 | 49.3 | 39.5 | 47.1±0.16 |
| ROID | 42.8 | 40.4 | 39.8 | 63.0 | 63.2 | 49.7 | 56.7 | 36.8 | 36.2 | 31.5 | 24.8 | 39.7 | 57.1 | 47.3 | 36.6 | 44.4±0.13 |
| +SSA | 43.0 | 40.8 | 39.8 | **62.8** | 64.2 | 49.5 | 55.5 | 36.8 | 35.7 | **31.1** | 24.4 | 39.8 | 54.8 | 45.6 | 35.3 | 43.9±0.11 |
| CMF | 42.9 | 40.3 | 39.8 | 64.0 | 63.4 | 49.5 | 55.1 | **36.7** | 34.9 | 32.4 | 23.4 | 38.4 | 52.0 | 44.9 | 35.6 | 43.5±0.04 |
| +SSA | **42.7** | **39.9** | **39.4** | 63.1 | **60.5** | **47.7** | **51.9** | 37.1 | **33.9** | **31.1** | **22.8** | **37.9** | **49.8** | **43.0** | **33.1** | **42.2±0.16** |

where $a > 0$ is a constant scalar. Consequently, the proposed algorithm seamlessly integrates with existing weight-based methods by inheriting the time-invariant covariance behavior described in Eq. (8).

By adopting a tiny initial step size $\eta$, the covariance $\mathbf{P}_{k+1}$ in Eq. (18) remains sufficiently small, ensuring that the posterior weight distribution is well-suited for the Bayesian weight enhancement framework. Within this framework, the posterior predictive distribution is simplified to $p(y|\boldsymbol{x}_{k+1}, \boldsymbol{m}_{k+1})$, enabling efficient inference without requiring sampling.

## 4. Experiments

### 4.1. Experimental Setup

**Datasets and Metrics** We evaluated our method using two widely recognized datasets, ImageNet-C (Hendrycks & Dietterich, 2019a) and D109 (Marsden et al., 2023), for distribution shifts in dynamic environments (Niu et al., 2023; Marsden et al., 2023). ImageNet (Deng et al., 2009) consists of 1,281,167 training samples and 50,000 testing samples. ImageNet-C extends ImageNet by applying 15 types of corruption at five severity levels. These corruptions are categorized into four groups: NOISE, BLUR, WEATHER, and DIGITAL. Consistent with prior studies (Niu et al., 2022; 2023; Marsden et al., 2023), we focused on the highest corruption level (level 5) to evaluate robustness against severe image degradation. The D109 dataset features five domains representing natural distribution shifts derived from DomainNet (Peng et al., 2019). It includes 109 classes that overlap with ImageNet, enabling cross-domain evaluations. The evaluation metrics were the mean and standard deviation of error rates computed across five random seeds to ensure robust performance measurement.

**Scenarios** For evolving distribution shifts over time, we leveraged several dynamic scenarios. In the covariate shift scenarios, sequentially arranged domains modeled time-correlated changes, where input samples were streamed domain by domain (Boudiaf et al., 2022; Wang et al., 2022; Yuan et al., 2023). The label shift scenarios were conducted by presenting sequential input samples drawn from specific label classes over time. Using a Dirichlet distribution parameter $\gamma$, we controlled the concentration of local label distributions; lower values $\gamma$ formed label-specific clusters, while $\gamma = \infty$ reproduced covariate shifts (Gong et al., 2022; Niu et al., 2023; Zhou et al., 2023). To evaluate the long-term stability of adaptation, we introduced periodic scenarios in which the domains followed a repeating sequence of covariate shifts. Each sequence was repeated up to 15 times to assess long-term adaptation behavior. In addition, we designed various order scenarios in which groups were presented in various orders to examine the order dependency of the TTA methods. For these, we created four group sequences (Order-1 to Order-4) to evaluate robustness under unpredictable distribution changes.

**Implementation Details** For all experiments, we used the base version of VisionTransformer (ViT) (Dosovitskiy et al., 2020) with the self-supervised data2vec (D2V) model (Baevski et al., 2022) as the backbone, consistent with previous works (Niu et al., 2023; Marsden et al., 2023; Lee & Chang, 2024). Source models were pre-trained on the ImageNet training dataset using publicly available weights to ensure reproducibility. Following previous studies (Li et al., 2018; Niu et al., 2022; 2023; Marsden et al., 2023; Lee & Chang, 2024), we limited the trainable weights to normalization layers, employing either batch normalization (Ioffe & Szegedy, 2015) or layer normalization (Ba et al., 2016), depending on the model architecture. Comparative methods were implemented using their official codebases and hyperparameters as described in their respective papers, ensuring alignment with the standard TTA benchmark (Marsden & Döbler, 2022). The models were trained with a batch size of 64 and a learning rate of 0.00001 using the

*Table 2.* Average error rates (%) and standard deviations in label shifts ($\gamma = 0.1$) on ImageNet-C. Red fonts indicate performance degradation with respect to Source.

| Method | NOISE | | | BLUR | | | | WEATHER | | | | DiGiTAL | | | | Avg. |
|---|---|---|---|---|---|---|---|---|---|---|---|---|---|---|---|---|
| | gaussian | shot | impulse | defocus | glass | motion | zoom | snow | frost | fog | bright | contrast | elastic | pixelate | jpeg | |
| Source | 43.9 | 43.3 | 43.4 | 69.7 | 78.3 | 59.6 | 69.1 | 40.1 | 44.3 | 36.3 | 26.5 | 50.6 | 67.6 | 60.6 | 43.4 | 51.8 |
| TENT | 44.0 | 43.5 | 43.8 | 70.8 | 78.3 | 59.9 | 68.8 | 42.4 | 52.0 | 56.5 | 30.2 | 64.7 | 68.7 | 63.2 | 44.7 | 55.4±1.58 |
| LAME | 45.7 | 43.7 | 45.2 | 72.4 | 88.0 | 60.2 | 87.5 | 89.1 | 95.0 | 99.7 | 27.1 | 95.7 | 95.0 | 63.2 | 44.2 | 70.1±0.04 |
| RoTTA | 43.5 | 41.2 | 40.8 | 68.4 | 71.1 | 56.3 | 64.4 | 39.1 | 38.3 | 38.6 | 28.3 | 65.4 | 67.5 | 67.4 | 49.4 | 52.0±0.06 |
| SAR | 43.9 | 41.7 | 40.9 | 68.4 | 71.8 | 55.0 | 63.4 | 39.3 | 39.1 | 38.8 | 25.3 | 44.8 | 58.0 | 49.9 | 39.3 | 48.0±0.10 |
| EATA | 43.5 | 40.5 | 39.6 | 61.7 | 62.0 | 48.1 | 56.0 | 36.7 | 36.0 | 32.9 | 23.0 | 37.1 | 53.2 | 44.5 | 34.7 | 43.3±0.03 |
| DeYO | 41.3 | 38.8 | 38.8 | 60.8 | 61.0 | 52.3 | 70.6 | 42.5 | 40.3 | 40.8 | 26.1 | 64.3 | 66.4 | 48.1 | 42.9 | 49.0±2.83 |
| ROID | 40.6 | 39.4 | 39.3 | 54.8 | 55.4 | 46.4 | 53.1 | 35.5 | 34.7 | 30.0 | 23.7 | 36.1 | 48.0 | 41.4 | 34.9 | 40.9±0.10 |
| +SSA | 40.5 | 38.8 | 38.7 | 54.3 | 53.8 | 45.0 | 51.9 | 34.7 | 34.1 | 29.5 | 23.0 | 35.7 | 45.1 | 39.5 | 33.2 | 39.9±0.04 |
| CMF | 40.3 | 38.5 | 38.4 | 52.7 | 49.9 | 42.5 | 46.4 | 34.0 | 32.8 | 28.6 | 22.2 | 34.4 | 42.1 | 38.6 | 31.6 | 38.2±0.05 |
| +SSA | **39.8** | **37.6** | **37.6** | **50.4** | **46.1** | **39.9** | **42.0** | **32.1** | **31.1** | **27.5** | **21.1** | **33.9** | **36.4** | **33.7** | **29.4** | **35.9±0.04** |

*Table 3.* Average error rates (%) and standard deviations in label shifts ($\gamma = 0.0$) on ImageNet-C. Red fonts indicate performance degradation with respect to Source.

| Method | NOISE | | | BLUR | | | | WEATHER | | | | DiGiTAL | | | | Avg. |
|---|---|---|---|---|---|---|---|---|---|---|---|---|---|---|---|---|
| | gaussian | shot | impulse | defocus | glass | motion | zoom | snow | frost | fog | bright | contrast | elastic | pixelate | jpeg | |
| Source | 43.9 | 43.3 | 43.4 | 69.7 | 78.3 | 59.6 | 69.1 | 40.1 | 44.3 | 36.3 | 26.5 | 50.6 | 67.6 | 60.6 | 43.4 | 51.8 |
| TENT | 44.1 | 43.7 | 44.0 | 71.1 | 79.2 | 61.6 | 69.8 | 43.2 | 53.1 | 55.9 | 30.8 | 48.7 | 69.4 | 69.1 | 58.9 | 56.2±0.98 |
| LAME | 30.5 | 29.8 | 30.2 | 49.4 | 62.4 | 39.9 | 50.3 | 31.3 | 34.3 | 31.4 | 22.7 | 39.9 | 55.9 | 41.4 | 34.5 | 38.9±0.07 |
| RoTTA | 43.8 | 42.0 | 42.0 | 69.9 | 74.5 | 59.3 | 67.4 | 40.3 | 39.5 | 40.2 | 29.0 | 74.5 | 72.4 | 72.8 | 51.5 | 54.6±0.04 |
| SAR | 44.2 | 41.8 | 41.0 | 67.6 | 71.7 | 54.8 | 63.5 | 39.2 | 39.0 | 38.2 | 25.6 | 67.5 | 66.0 | 57.9 | 39.0 | 50.5±1.38 |
| EATA | 44.2 | 41.4 | 40.8 | 64.7 | 66.7 | 52.2 | 60.5 | 39.7 | 40.6 | 39.4 | 24.8 | 46.0 | 55.4 | 49.7 | 38.3 | 47.0±0.11 |
| DeYO | 41.5 | 38.9 | 38.9 | 61.7 | 61.3 | 51.8 | 72.0 | 42.2 | 41.6 | 39.7 | 26.5 | 56.4 | 57.1 | 47.3 | 41.4 | 47.9±0.57 |
| ROID | **12.2** | 11.8 | 11.6 | 32.5 | 33.5 | 18.4 | 30.1 | 12.4 | 11.6 | 9.8 | 7.3 | 12.6 | 25.1 | 15.3 | 13.0 | 17.1±0.32 |
| +SSA | **12.2** | 11.8 | 11.5 | 32.6 | 29.6 | 18.3 | 28.9 | 12.1 | 11.3 | 9.6 | 7.1 | 12.1 | 23.2 | 14.1 | 11.5 | 16.4±0.14 |
| CMF | 12.4 | 12.0 | 11.9 | **28.8** | 23.9 | 15.6 | 22.4 | 11.2 | 10.2 | 8.9 | 6.3 | 11.3 | 17.9 | 13.0 | 9.7 | 14.4±0.24 |
| +SSA | 12.3 | **11.4** | **11.3** | 29.2 | **20.5** | **14.5** | **19.4** | **10.5** | **9.7** | **8.3** | **6.1** | **10.0** | **14.6** | **10.8** | **8.7** | **13.1±0.29** |

*Table 4.* Average error rates (%) comparison with various methods in the periodic scenario.

| Round | | | | Method | | |
|---|---|---|---|---|---|---|
| | Source | EATA | DeYO | ROID | CMF | SSA |
| 1 | | 50.2 | 47.1 | 44.4 | 43.3 | **42.2** |
| 2 | | 46.6 | 45.6 | 44.3 | 42.0 | **40.8** |
| 3 | | 45.1 | 45.6 | 44.3 | 41.7 | **40.6** |
| 4 | | 44.2 | 45.7 | 44.3 | 41.6 | **40.4** |
| 5 | | 43.7 | 47.2 | 44.3 | 41.6 | **40.2** |
| 6 | | 43.3 | 47.4 | 44.3 | 41.5 | **40.2** |
| 7 | | 43.1 | 48.6 | 44.3 | 41.5 | **40.4** |
| 8 | 51.8 | 42.9 | 49.3 | 44.3 | 41.5 | **40.2** |
| 9 | | 42.7 | 50.9 | 44.3 | 41.4 | **40.3** |
| 10 | | 42.7 | 54.3 | 44.4 | 41.4 | **40.4** |
| 11 | | 42.6 | 57.2 | 44.3 | 41.4 | **40.3** |
| 12 | | 42.5 | 54.3 | 44.3 | 41.4 | **40.4** |
| 13 | | 42.4 | 55.2 | 44.4 | 41.4 | **40.2** |
| 14 | | 42.3 | 57.0 | 44.3 | 41.4 | **40.4** |
| 15 | | 42.3 | 60.0 | 44.3 | 41.4 | **40.3** |
| Avg. | 51.8 | 43.8±2.14 | 51.0±4.88 | 44.3±0.03 | 41.6±0.50 | **40.5±0.51** |

SGD optimizer. By default, SSA was integrated with CMF, setting the hyperparameter $a = 0.01$, following the ROID and CMF configurations. The steady-state scale factor $\sigma_\lambda^2$ was set to $10^{-12}$.

### 4.2. Effectiveness

The performance of TTA methods in the covariate shift scenario on ImageNet-C is summarized in Table 1. Weight-based TTA methods, particularly ROID and CMF, demonstrated significantly better performance compared to other approaches. This result supports our hypothesis that leveraging weight distributions and incorporating multiple weights with consistent mixing ratios enhances robustness in dynamic environments. By incorporating the proposed SSA algorithm, these methods achieved state-of-the-art performance in all domains, except the snow domain. Tables 2 and 3 present the performance of the TTA methods in the label shift scenario, with the intensities set to $\gamma = 0.1$ and $\gamma = 0.0$, respectively. In both cases, SSA consistently improved the average performance of ROID and CMF, highlighting its effectiveness in addressing significant label imbalances. Similar trends were observed on the D109 dataset (detailed in Appendix B), where SSA consistently enhanced the robustness of weight-based methods. These results underscore the generalizability of SSA to different datasets and adaptation scenarios.

### 4.3. Stability

Table 4 presents the performance of various TTA methods in the periodic scenario as the number of rounds increases. Initially, EATA lags in performance but improves rapidly over successive rounds, eventually surpassing ROID. However, CMF consistently outperforms both EATA and ROID in all rounds, achieving higher average performance. By integrating SSA, CMF further improves, maintaining supe-

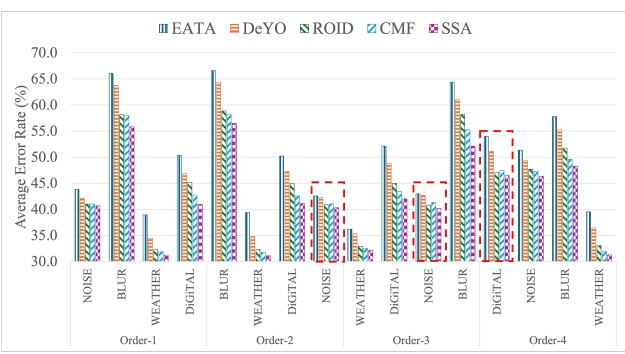

*Figure 2.* Average error rates (%) comparison with various methods in various order scenario.

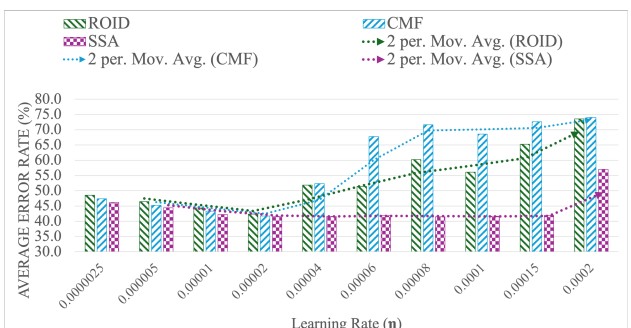

*Figure 3.* Average error rates (%) of different methods as learning rate increase.

rior performance throughout the adaptation process. These results highlight the ability of SSA to balance high adaptability with long-term stability, making it a reliable solution for scenarios that require sustained performance.

### 4.4. Adaptability

Figure 2 illustrates the performance of TTA methods in four different group orders. While CMF generally outperforms ROID, it exhibits vulnerabilities in certain cases. Specifically, CMF underperforms in the NOISE domain for Order-2 and Order-3 and shows degraded performance in the DIGITAL domain for Order-4, as indicated by the red dotted boxes. By integrating SSA, these limitations are effectively mitigated, with SSA achieving the highest performance across all groups and orders compared to existing methods. These results demonstrate the adaptability of SSA to unpredictable distribution shifts, reinforcing its robustness and versatility in handling diverse and dynamic environments.

## 5. Analysis

We conducted an in-depth analysis of sensitivity to learning rates, computational efficiency, and the role of covariance in driving its performance. These analyses provide insights into the core mechanisms and effectiveness of SSA.

**Learning Rate Sensitivity**  A common challenge for TTA methods is the instability caused by increasing

*Table 5.* Average error rates (%) and elapsed time comparison of methods with and without SSA in covariate shift.

| Method | Time | Relative Time (%) | Avg. |
|---|---|---|---|
| TENT | 1565.2 | 0.0 | 53.3±0.22 |
| TENT+SSA | 1580.7 | 1.0 | **49.3±0.30** |
| ROID | 1732.3 | 0.0 | 44.4±0.13 |
| ROID+SSA | 1790.4 | 3.4 | **43.9±0.11** |
| CMF | 1751.4 | 0.0 | 43.5±0.04 |
| CMF+SSA | 1827.4 | 4.3 | **42.2±0.16** |

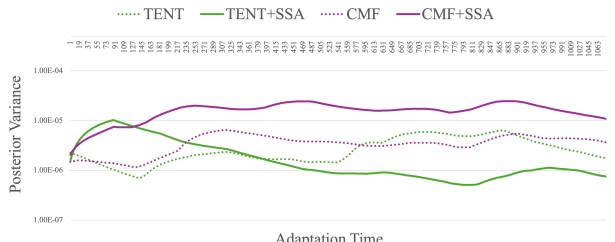

*Figure 4.* Comparison of variance evolution with and without SSA.

learning rates (Zhao et al., 2023). As shown in Figure 3, when the learning rate exceeded 0.00004, both ROID and CMF experienced significant performance degradation. In contrast, SSA effectively mitigated this instability, maintaining stable performance even at learning rates up to 15 times the default value. Beyond this threshold, SSA's performance also declined due to violations of the small learning rate assumption inherent to the SDE approximation. Despite this limitation, SSA significantly improved the stability of existing methods across a wide range of learning rates.

**Efficiency**  Efficiency is a essential consideration for TTA methods, as they operate in an online learning setting where computational overhead must be minimized. Table 5 reports the elapsed time and relative time for various TTA methods under the covariate shift scenario. SSA-enhanced methods exhibited minimal relative time increases of 1.0%, 3.4%, and 4.3% compared to their respective baselines, while delivering significant performance improvements. Notably, SSA boosted TENT's performance by 4.0%, effectively mitigating weight degradation with minimal additional computational cost. These results highlight SSA's efficiency, as it enhances adaptation by refining the existing trajectory of weight updates without requiring additional optimization steps.

**Covariance Profile**  The core mechanism of SSA lies in balancing covariance. Figure 4 illustrates the evolution of the scalar variance $p_k$ of the posterior covariance $\mathbf{P}_k$ over the adaptation process in the covariate shift scenario on ImageNet-C. Throughout the process, the variance remained consistently within the range of $10^{-7}$ to $10^{-4}$, demonstrating its consistently small magnitude. This result supports the theoretical assertion of our framework that a small variance must be maintained. For TENT, SSA effectively reduced both the variance and its fluctuations, leading to a stabilized adaptation process.

In the case of CMF, SSA slightly increased the variance, but simultaneously controlled its variability, thereby enhancing the stability. These results empirically validate the hypothesis that covariance is a critical factor influencing TTA performance and demonstrate the effectiveness of SSA in dynamically balancing covariance.

## 6. Related Works

**Sample-based Test-time Adaptation** TTA is an online learning paradigm that adapts a pre-trained source model to unlabeled test samples drawn from a different distribution. Many TTA methods rely on entropy minimization, using model outputs as target probability to facilitate unsupervised adaptation (Wang et al., 2020). However, in dynamic environments characterized by diverse changing distribution shifts, these methods are prone to model collapse (Boudiaf et al., 2022; Gong et al., 2022; Niu et al., 2023; Press et al., 2024). Model collapse typically arises from explicit gradient noise, which is introduced by unreliable model predictions, leading to weight degradation over time. To mitigate these issues, sample-based TTA methods reduce the influence of test samples associated with poor predictions (Niu et al., 2022; Lee et al., 2024; Marsden et al., 2023). These methods attempt to filter out unreliable samples by selecting high-confidence samples during adaptation. However, their dependence on unreliable model output, which becomes increasingly corrupt under extreme distribution shifts, limits their overall effectiveness. As a result, these methods remain vulnerable to performance degradation in highly dynamic or extreme shifts. In contrast, weight-based TTA methods (Niu et al., 2022; 2023; Marsden et al., 2023; Lee & Chang, 2024) alleviate this problem by continuously averaging the source weight with the current weight. This approach addresses weight degradation and integrates naturally into our proposed Bayesian weight enhancement framework.

**Bayesian Deep Learning** Highly flexible models such as DNNs can represent a wide range of functions, each with different generalization properties. Considering these multiple models improves accuracy in new data distributions. This concept is often implemented through Bayesian model averaging, where model weights are treated as samples drawn from a weight distribution (Polson & Sokolov, 2017; Wilson, 2020; Wilson & Izmailov, 2020; Khan, 2019). The weight distribution is commonly approximated as a Gaussian distribution centered around local modes (Chaudhari & Soatto, 2018). Since weights obtained via SGD tend to cluster around regions with good generalization properties, averaging multiple SGD samples collected at regular intervals improves robustness to distribution shifts (Izmailov et al., 2018; Garipov et al., 2018; Madry et al., 2017). These approaches have

empirically demonstrated robustness to out-of-distribution data, particularly in fine-tuning large foundation models (Wortsman et al., 2022; Rame et al., 2022). However, the study of time-dependent weight distributions under unsupervised learning, where explicit gradient noise can arise, remains an underexplored area. Our research addresses this gap by theoretically deriving the weight distribution using SDE approximations and Bayesian filtering and further demonstrating its empirical effectiveness in improving model performance.

**Bayesian Filtering** Bayesian filtering is a recursive Bayesian inference method for predicting and updating time-evolving observations. Kalman filtering, in particular, performs exact Bayesian inference under the assumption of linear Gaussian models (Cheng et al., 2019; Abuduweili & Liu, 2020). For nonlinear observations, extensions such as Extended Kalman Filtering (EKF) and Iterated EKF have been developed (Kloeden et al., 1992; Bell & Cathey, 1993). These techniques have been applied to DNN outputs, where Kalman filtering is performed with linearized approximations (Puskorius & Feldkamp, 2001). Our approach shares this idea by using linearization from Eq. (24) and substituting local gradient and covariance terms with observed constants. The validity of this substitution is confirmed through experiments in various scenarios, datasets, and models. Another Bayesian inference approach, particle filtering, can also capture complex weight distributions (Huang et al., 2022). However, particle filtering requires sampling weights, which introduces significant computational overhead. In contrast, SSA directly leverages the mean of the posterior weight distribution, leveraging the small-variance characteristic of TTA processes. This SSA behavior ensures computational efficiency, making SSA a practical and scalable solution for TTA.

## 7. Conclusion

In this paper, we addressed the weight degradation problem in the TTA process caused by explicit gradient noise and introduced the Bayesian weight enhancement framework, which generalizes existing weight-based TTA methods effective in mitigating the problem. Building on this probabilistic framework, we identified a key limitation in weight-based approaches: their neglect of time-varying covariance, which captures changes in gradient noise. To address this, we theoretically derived the covariance using the SDE approximation and Bayesian inference, leading to the development of the SSA algorithm. The algorithm consistently improved the performance of the state-of-the-art TTA method by dynamically balancing covariance across diverse datasets, scenarios, and model architectures. These results highlight the covariance dynamics that drive TTA performance in dynamic environments.

## Acknowledgements

This work was supported by Hankuk University of Foreign Studies Research Fund (Of 2025)

## Impact Statement

This paper presents work whose goal is to advance the field of Machine Learning. There are many potential societal consequences of our work, none which we feel must be specifically highlighted here.

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

# A. Technical Details and Derivation

In this section, we discuss the integration of SSA and SSA into a unified algorithm. In addition, we examine the validity of the SDE approximation and provide a detailed derivation of the SSA.

## A.1. Continual Momentum Filtering with SSA

---

**Algorithm 2** Continual Momentum Filtering with Steady-State Adaptation

---

**Require:** Learning rate $\eta$, Source model $f(.; \hat{w}_0)$
**Require for SSA:** $\sigma_\lambda^2$, $A = a\mathbf{I}$
**Require for CMF:** $\beta$, $\mathbf{S} = s\mathbf{I}$
Initialization $\mathbf{m}_0 = \hat{w}_0, \mathbf{P}_0 = \mathbf{0}, \hat{g}_0 = 0, \boldsymbol{h}_0 = \hat{w}_0, \mathbf{Q}_0 = \mathbf{0}$
**for** $k = 1$ **to** $K$ **do**
  $g_k \leftarrow \nabla_{\boldsymbol{m}_k} G(\boldsymbol{m}_k)$

  **Steady-state adaptation:**
  $\bar{g}_k \leftarrow \frac{1}{k}(g_k + \hat{g}_{k-1})$
  $\sigma_k^2 \leftarrow \frac{1}{d}\text{tr}\Big((g_k - \bar{g}_k)(g_k - \bar{g}_k)^\top\Big)$
  $\alpha_k \leftarrow \sqrt{\sigma_\lambda^2/(\eta^2 \sigma_k^2)}$
  $\boldsymbol{m}_{k+1|k}^+ \leftarrow \boldsymbol{m}_k - \alpha_k \eta g_k$
  $\mathbf{P}_{k+1|k}^+ \leftarrow \mathbf{P}_k + \alpha_k^2 \eta^2 \Sigma_k$
  $\hat{g}_k \leftarrow (\alpha_k g_k + \hat{g}_{k-1})$

  **Continual momentum filtering:**
  $\boldsymbol{h}_{k+1|k}^+ \leftarrow \beta \boldsymbol{h}_k + (1-\beta)\hat{w}_0$
  $\mathbf{Q}_{k+1|k}^+ \leftarrow \beta^2 \mathbf{Q}_k + \mathbf{S}$
  $B_t = \mathbf{Q}_{t|t-1}^+ (\beta^2 \mathbf{Q}_{t-1|t-1}^+ + 1)^{-1}$
  $\boldsymbol{h}_{k+1} \leftarrow \boldsymbol{h}_{k+1|k}^+ + B_t(\boldsymbol{m}_{k+1|k}^+ - \boldsymbol{h}_{k+1|k}^+)$
  $\mathbf{Q}_{k+1} \leftarrow (\mathbf{I} - K_t)\mathbf{Q}_{k+1|k}^+$

  **Bayesian weight enhancement framework :**
  $\boldsymbol{m}_{k+1} \leftarrow \boldsymbol{m}_{k+1|k}^+ + A(\boldsymbol{h}_{k+1} - \boldsymbol{m}_{k+1|k}^+)$
  $\mathbf{P}_{k+1} \leftarrow (\mathbf{I} - A)\mathbf{P}_{k+1|k}^+$
**end for**

---

Algorithm 2 details the integration of SSA with CMF forming a unified TTA algorithm. SSA achieves state-of-the-art performance when combined with CMF in various scenarios by effectively regulating weight dynamics. CMF utilizes Kalman filtering to update the source model with the current weight, resulting in the hidden source weight distribution $p(\boldsymbol{w}_{k+1}|\boldsymbol{u}_{k+1})$. In contrast, SSA estimates $p(\boldsymbol{u}_{k+1}|\boldsymbol{w}_{0:k})$ using the transition weight distribution derived from SDE and the Chapman-Kolmogorov equation in Eq. (13). These two distributions are naturally integrated into Eq. (16), where SSA replaces the fixed source weight $\hat{w}_0$ with the hidden source weight $\boldsymbol{h}_{t+1}$ obtained through CMF. This integration seamlessly combines two different methods, allowing SSA to take advantage of the flexible source weight derived from CMF for improved adaptability.

## A.2. Validity of SDE Approximation in TTA Process

The validity of the SDE approximation for SGD arises from the requirement that the learning rate $\eta$ be sufficiently small (Zhu et al., 2018; Li et al., 2021). In the TTA process, the connection between discrete time steps and continuous time is defined as $t = k\eta$, where $k$ denotes the iteration step. This mapping ensures that discrete updates in TTA can reasonably be modeled as a continuous stochastic process when $\eta$ is small. TTA methods typically update less than 1% of a model's total weights (Yuan et al., 2023; Niu et al., 2022; Marsden et al., 2023). This TTA setting indicates that the learning rates for 99% weight are zero. Furthermore, TTA methods adopt learning rates in the range of $10^{-5}$ to $10^{-6}$, which are approximately 100

times smaller than the learning rates commonly used for training source models (i.e., $10^{-3}$ to $10^{-4}$) (Steiner et al., 2021; Wang et al., 2020; 2022). This nature of low learning rates across TTA methods ensures that the gradient noise introduced by SGD remains manageable and aligns with the assumptions of the SDE approximation. Consequently, the small learning rates widely adopted in TTA provide strong empirical and theoretical support for the validity of the SDE approximation in this context.

### A.3. Transition Distribution from SDE Approximation

In this section, we derive the transition weight distribution from the SDE approximation of the TTA process. The approximation for SGD in the TTA process is expressed as:

$$d\boldsymbol{u}_t = -g(\boldsymbol{u}, t)dt + \sqrt{\eta}\Sigma(\boldsymbol{u}, t)^{1/2}dW_t, \tag{22}$$

where $g(\boldsymbol{w}, t)$ and $\Sigma(\boldsymbol{w}, t)$ are $g_t$ and $\Sigma_t = \sigma_t\mathbf{I}$ in Eq. (9). For this SDE, the evolution of the weight distribution $p(\boldsymbol{u}_t)$ is governed by following the FPK equation:

$$\frac{\partial p(\boldsymbol{u}_t)}{\partial t} = \sum_{i=1}^{d}\frac{\partial p(\boldsymbol{u}_t)}{\partial w_t^i}[g(\boldsymbol{u}, t)]_i + \frac{1}{2}\sum_{i=1}^{d}\sum_{j=1}^{d}\frac{\partial^2 p(\boldsymbol{u}_t)}{\partial w_t^i \partial w_t^j}\eta[\Sigma(\boldsymbol{u}, t)]_{ij}, \tag{23}$$

where $[\cdot]_i$ and $[\cdot]_{ij}$ represent the $i$-th element of a vector $[\cdot]$, and $(i, j)$-th element of a matrix $[\cdot]$, respectively. Since $g(\boldsymbol{u}, t)$ and $\Sigma(\boldsymbol{u}, t)$ are generally intractable, we approximate $p(\boldsymbol{u}_t)$ as a Gaussian distribution $\mathcal{N}(\boldsymbol{u}_t|\boldsymbol{m}, \mathbf{P})$ following the Gaussian assumed density approximation from (Särkkä & Solin, 2019). However, this approximation requires the efficient computation of an n-dimensional Gaussian integral. Following Theorem 9.2 and Algorithm 9.4 from (Särkkä & Solin, 2019), we linearize (via Taylor series) the gradient $g(\boldsymbol{m}, t)$ and approximate the $\sqrt{\eta}\Sigma(\boldsymbol{u}, t)^{1/2}$ around the mean $\boldsymbol{m}$ as:

$$g(\boldsymbol{u}, t) \approx g(\boldsymbol{m}, t) + \mathcal{G}(\boldsymbol{u}, t)(\boldsymbol{u} - \boldsymbol{m}), \quad \sqrt{\eta}\Sigma(\boldsymbol{u}, t)^{1/2} \approx \sqrt{\eta}\Sigma(\boldsymbol{m}, t)^{1/2}, \tag{24}$$

where $\mathcal{G}(\boldsymbol{u}, t)$ represents the Jacobian matrix of $g(\boldsymbol{u}, t)$ with respect to $\boldsymbol{u}$. Thus, the mean and covariance are derived as follows:

$$\frac{d\boldsymbol{m}}{dt} = -g(\boldsymbol{w}, t), \quad \frac{d\mathbf{P}}{dt} = \mathbf{P}\mathcal{G}_t^\top + \mathcal{G}_t\mathbf{P} + \eta\Sigma(\boldsymbol{u}, t). \tag{25}$$

In this context, the transition weight distribution can be constructed using a linear Gaussian model (Särkkä & Solin, 2019). For arbitrary intervals $0 < s < t$ with initial conditions $\boldsymbol{m}_s = \hat{\boldsymbol{w}}_k, \mathbf{P}_s = \mathbf{0}$, the transition distribution is given by $p(\boldsymbol{u}_t|\boldsymbol{u}_s) \approx \mathcal{N}(\boldsymbol{u}_{k+1}|\boldsymbol{m}_{t|s}, \mathbf{P}_{t|s})$. For the discrete interval $s = k\eta$ and $t = (k+1)\eta$, the discrete transition distribution $p(\boldsymbol{u}_{k+1}|\boldsymbol{u}_k)$ simplifies to $\mathcal{N}(\boldsymbol{u}_{k+1}|\boldsymbol{m}_{k+1|k}, \mathbf{P}_{k+1|k})$. Over this small interval, $g(\boldsymbol{u}, t)$ and $\Sigma(\boldsymbol{u}, t)$ can be constant, denoted as $g_k = \nabla G(\hat{\boldsymbol{u}}_t)$ and $\sigma_k^2 = \frac{1}{d}\text{tr}\left((g_k - \bar{g}_k)(g_k - \bar{g}_k)^\top\right)$, where $\mathcal{G}_t$ is approximated as $\mathbf{0}$. Consequently, Eq. (25) is reduced to linear ordinary differential equations, yielding the mean and covariance at discrete time $k$:

$$\boldsymbol{m}_{k+1|k} = \hat{\boldsymbol{w}}_k - \int_s^t g_k dt = \hat{\boldsymbol{w}}_k - g_k\Delta t, \quad \mathbf{P}_{k+1|k} = \int_s^t \eta\sigma_k^2\mathbf{I} = \sigma_k^2\Delta t^2\mathbf{I}, \tag{26}$$

where $\Delta t = \eta$. This transition distribution is combined with Bayesian filtering to derive the posterior weight distribution in Eq. (16).

### A.4. Balancing Covariance

To maintain the covariance of the weight posterior distribution near the steady-state, it is necessary to satisfy $\mathbf{P}_{k+1} \approx \mathbf{P}_k$. Since all covariance matrices $\mathbf{P}_{k+1}$ and $\mathbf{P}_k$ in Eqs. (15) and (18) are scalar multiples of the identity matrix, we can consider only the scalar term (i.e., variance) in $\mathbf{P}_k = \text{p}_k\mathbf{I}$, $\Sigma_k = \sigma_k^2\mathbf{I}$ and $R = \sigma_r^2\mathbf{I}$. At steady-state, the variance no longer changes between steps:

$$\text{p}_k = \text{p}_{k+1} = \text{p}_\infty. \tag{27}$$

Substituting this into the scalar forms of Eqs. (15) and (18), we have:

$$\begin{aligned}
\text{p}_\infty &= (\text{p}_\infty + \sigma_k^2\Delta t^2)\left(1 - \frac{\text{p}_\infty + \sigma_k^2\Delta t^2}{\text{p}_\infty + \sigma_k^2\Delta t^2 + \sigma_r^2}\right) \\
&= (\text{p}_\infty + \sigma_k^2\Delta t^2)\frac{\sigma_r^2}{\text{p}_\infty + \sigma_k^2\Delta t^2 + \sigma_r^2}
\end{aligned} \tag{28}$$

Rearranging this equation yields:

$$\mathrm{p}_\infty(\mathrm{p}_\infty + \sigma_k^2 \Delta t^2 + \sigma_r^2) = \sigma_r^2(\mathrm{p}_\infty + \sigma_k^2 \Delta t),$$

$$\mathrm{p}_\infty^2 + \mathrm{p}_\infty \sigma_k^2 \Delta t^2 + \mathrm{p}_\infty \sigma_r^2 - \mathrm{p}_\infty \sigma_r^2 - \sigma_k^2 \sigma_r^2 \Delta t^2 = 0 \tag{29}$$

$$\mathrm{p}_\infty^2 - \sigma_k^2(\sigma_r^2 - \mathrm{p}_\infty)\Delta t^2 = 0$$

From this, the step size $\Delta t$ is derived as:

$$\Delta t = \sqrt{\frac{\mathrm{p}_\infty}{\sigma_k^2(\sigma_r^2 - \mathrm{p}_\infty)}}, \tag{30}$$

where $\mathrm{p}_\infty < \sigma_r^2$. To define the dynamic step size $\Delta t_k = \alpha_k \eta$ where $\eta$ is the base learning rate, we rewrite $\alpha_k$ as:

$$\alpha_k = \sqrt{\frac{\sigma_\lambda^2}{\eta^2 \sigma_k^2}}. \tag{31}$$

where $\sigma_\lambda = \sqrt{\mathrm{p}_\infty/(\sigma_r^2 - \mathrm{p}_\infty)}$. This result is substituted into $\Delta t$ in Eqs. (14) and (15), ensuring that the posterior weight distribution in Eq. (16) maintains the steady-state condition, thus balancing covariance. Importantly, under the steady-state condition, $A_k$ converges to $A = a\mathbf{I}$ with a fixed scalar $a > 0$, representing time-invariant covariance. Consequently, the proposed algorithm integrates naturally with weight-based TTA methods described in Eq. (7), maintaining compatibility while improving stability and adaptability.

## B. Experiments Details

### B.1. Experimental Setup

Our experiments were conducted using a single NVIDIA GeForce RTX 3090 GPU. This section outlines the specific experimental settings used in our study. To ensure robustness and reproducibility, the evaluation metrics include the mean and standard deviation of error rates across five random seeds (1, 2, 3, 4, and 5).

**Datasets** The datasets used in our experiments were selected to evaluate the effect of diverse classes, corruption types, and natural distribution shifts commonly encountered in real-world scenarios. ImageNet-C was employed as a standard benchmark for evaluating robustness against corruption. ImageNet consists of 1,281,167 training samples and 50,000 testing samples, while ImageNet-C extends ImageNet by applying 15 types of corruption (e.g., Gaussian noise, shot noise, defocus blur, frost, and JPEG compression) at five severity levels. Consistent with previous studies (Boudiaf et al., 2022; Niu et al., 2022; 2023; Marsden et al., 2023; Lee & Chang, 2024), we used severity level 5, treating each type of corruption as a distinct domain. To further assess model performance under natural distribution shifts, we utilized D109, derived from DomainNet. D109 includes five domains (clipart, infograph, painting, real, and sketch) and comprises 109 classes that overlap with ImageNet. For an additional exploration of natural distribution shifts, we incorporated the Rendition and Sketch datasets in the covariate shift scenario. Rendition contains 30,000 images of 200 ImageNet classes rendered in artistic styles, collected from Flickr and curated via Amazon Mechanical Turk. The Sketch dataset consists of 50,000 black-and-white images, with 50 sketches for each of the 1,000 ImageNet classes, constructed using Google image queries.

**Compared Methods** We compared SSA with several state-of-the-art TTA methods, each employing different strategies to improve adaptation to distribution shifts. TENT (Wang et al., 2020) updates trainable weights using entropy minimization loss, allowing the model to adjust its predictions to reduce uncertainty in test samples. LAME (Boudiaf et al., 2022) adapts to label distribution shifts by modifying model outputs rather than updating the model weights, ensuring stable adaptation without direct changes of parameters. RoTTA (Yuan et al., 2023) employs a student-teacher approach with cross-entropy objectives and data augmentation to improve robustness against shifting distributions. EATA (Niu et al., 2022) follows an entropy-based objective while excluding high-entropy samples based on a predefined threshold, preventing noisy gradient updates from degrading model performance. SAR (Niu et al., 2023) integrates sample exclusion with sharpness-aware minimization to avoid sharp local optima. Additionally, SAR monitors the model loss and resets the model to its source state if the loss exceeds a predefined threshold. DeYO (Lee et al., 2024) employs a sample selection strategy based on pseudo-label probability differences and entropy. It identifies high-confidence samples by applying object-destructive transformations and measuring prediction changes. ROID (Marsden et al., 2023) incorporates an entropy objective that accounts for label distribution diversity, while

excluding low-confidence samples during training. Additionally, ROID continuously averages the source weight with the current weight to enhance stability during adaptation. CMF (Lee & Chang, 2024) uses Kalman filtering to dynamically update the source weight by mixing it with the current weight. The updated source weight is then further integrated with the target weight, ensuring smooth adaptation while preserving knowledge from the pre-trained model.

**Implementation Details**  Our experiments were implemented using the base version of ViT (Dosovitskiy et al., 2020) with the self-supervised D2V model (Baevski et al., 2022) as the backbone, following previous work (Niu et al., 2023; Marsden et al., 2023; Lee & Chang, 2024). In addition, we used SwinTransformer (Liu et al., 2021) as an alternative architecture. All source models were pre-trained on the ImageNet training dataset using publicly available weights to ensure reproducibility. For all experiments, we adhered to the hyperparameters specified in the TTA benchmark (Marsden & Döbler, 2022) and followed the official implementations and values reported in the original papers of each method. If hyperparameters were unavailable for a specific dataset or model, we adjusted them accordingly. All experiments used the SGD optimizer with a momentum of 0.9. The learning rates were set as follows: 0.00001 for D2V, 0.00025 for ViT, and 0.000005 for EATA. Swin used the same learning rate as ViT, while SAR employed the SAM optimizer with a learning rate of 0.001 for both ViT and Swin. For CMF, we used $(\beta, s) = (0.99, 0.005)$. Following ROID and CMF, we set the SSA hyperparameter $a = 0.01$. The steady-state scale factor $\sigma_\lambda^2$ was set to $10^{-12}$, and SSA was applied once $\eta^2 \sigma_k^2$ reached half of this factor.

## B.2. Additional Experiments

*Table 6.* Average error rates (%) and standard deviations in covariate shift ($\gamma = \infty$) on D109. Red fonts indicate performance degradation with respect to Source.

| Method | Adaptation Order ($\rightarrow$) | | | | | Avg. |
| --- | --- | --- | --- | --- | --- | --- |
| | clipart | infograph | painting | real | sketch | |
| Source | 48.7 | 72.9 | 41.2 | 20.5 | 56.7 | 48.0 |
| TENT | 49.1 | 77.5 | 51.4 | 31.2 | 79.4 | 57.7±0.08 |
| LAME | 98.7 | 99.6 | 96.4 | 51.3 | 99.1 | 89.0±0.14 |
| RoTTA | 48.6 | 72.6 | 40.7 | 19.9 | 53.9 | 47.2±0.01 |
| SAR | 48.3 | 74.4 | 42.9 | 20.3 | 56.5 | 48.5±0.10 |
| EATA | 47.9 | 71.6 | 40.0 | 19.7 | 54.1 | 46.6±0.05 |
| DeYo | 47.2 | 74.5 | 41.3 | 19.8 | 51.3 | 46.8±0.56 |
| ROID | 43.5 | 68.5 | 37.7 | 19.3 | 50.4 | 43.9±0.04 |
| +SSA | 43.7 | 67.0 | 36.9 | 18.8 | 47.9 | 42.9±0.06 |
| CMF | 43.0 | 66.5 | 36.3 | 18.5 | 47.3 | 42.3±0.11 |
| +SSA | **42.1** | **65.3** | **35.8** | **18.2** | **46.0** | **41.5±0.06** |

*Table 7.* Average error rates (%) and standard deviations in label shifts ($\gamma = 0.1$) on D109. Red fonts indicate performance degradation with respect to Source.

| Method | Adaptation Order ($\rightarrow$) | | | | | Avg. |
| --- | --- | --- | --- | --- | --- | --- |
| | clipart | infograph | painting | real | sketch | |
| Source | 48.7 | 72.9 | 41.2 | 20.5 | 56.7 | 48.0 |
| TENT | 49.1 | 77.4 | 51.3 | 31.7 | 79.7 | 57.8±0.06 |
| LAME | 69.8 | 94.5 | 57.6 | 32.7 | 68.3 | 64.6±0.25 |
| RoTTA | 48.7 | 72.7 | 40.9 | 20.2 | 55.4 | 47.6±0.03 |
| SAR | 48.4 | 74.6 | 43.5 | 20.3 | 56.4 | 48.6±0.02 |
| EATA | 47.8 | 71.5 | 39.9 | 19.8 | 53.7 | 46.5±0.06 |
| DeYo | 47.3 | 74.4 | 40.6 | 19.7 | 51.0 | 46.6±0.40 |
| ROID | 35.0 | 63.1 | 28.1 | 12.9 | 41.0 | 36.0±0.08 |
| +SSA | 35.3 | 61.3 | 27.4 | 12.6 | 38.1 | 34.9±0.07 |
| CMF | 34.4 | 60.3 | 26.6 | 12.1 | 36.8 | 34.1±0.13 |
| +SSA | **33.7** | **59.2** | **26.4** | **12.0** | **35.5** | **33.4±0.06** |

*Table 8.* Average error rates (%) and standard deviations in label shifts ($\gamma = 0.0$) on D109. Red fonts indicate performance degradation with respect to Source.

| Method | Adaptation Order ($\rightarrow$) | | | | | Avg. |
| --- | --- | --- | --- | --- | --- | --- |
| | clipart | infograph | painting | real | sketch | |
| Source | 48.7 | 72.9 | 41.2 | 20.5 | 56.7 | 48.0 |
| TENT | 49.2 | 77.1 | 51.5 | 32.6 | 80.9 | 58.2±0.04 |
| LAME | 26.0 | 68.8 | **19.2** | **8.0** | **26.7** | 29.7±0.15 |
| RoTTA | 48.7 | 72.9 | 41.1 | 20.5 | 56.7 | 48.0±0.01 |
| SAR | 48.7 | 75.4 | 46.8 | 20.1 | 56.6 | 49.5±0.05 |
| EATA | 47.9 | 71.5 | 39.6 | 19.7 | 54.0 | 46.5±0.14 |
| DeYo | 48.9 | 75.8 | 45.1 | 20.7 | 52.2 | 48.5±0.39 |
| ROID | 25.3 | 55.5 | 21.6 | 10.5 | 33.1 | 29.2±0.04 |
| +SSA | 25.4 | 52.9 | 21.7 | 10.5 | 30.7 | 28.3±0.05 |
| CMF | 24.9 | 52.8 | 20.4 | 10.2 | 30.9 | 27.8±0.12 |
| +SSA | **24.4** | **51.3** | 20.4 | 10.0 | 30.6 | **27.4±0.21** |

*Table 9.* Average error rates (%) and standard deviations on Rendition and Sketch. Red fonts indicate performance degradation with respect to Source.

| Method | Rendition | Sketch |
| --- | --- | --- |
| Source | 46.6 | 60.4 |
| TENT | 46.0±0.03 | 60.3±0.06 |
| LAME | 86.7±0.39 | 86.9±0.37 |
| RoTTA | 46.5±0.01 | 60.1±0.03 |
| SAR | 45.9±0.05 | 60.2±0.07 |
| EATA | 45.8±0.09 | 58.6±0.08 |
| DeYo | 42.9±0.07 | 60.4±0.62 |
| ROID | 41.4±0.08 | 55.7±0.02 |
| +SSA | 40.3±0.13 | 54.5±0.07 |
| CMF | 39.7±0.11 | 53.3±0.06 |
| +SSA | **38.8±0.12** | **52.5±0.07** |

Tables 6, 7, and 8 present results omitted from Section 4.2. SSA consistently improved ROID and CMF performance in all scenarios in the D109 dataset, demonstrating its robustness in handling natural distribution shifts. We further evaluated TTA methods under a prolonged long one-domain natural shift scenario. Table 9 reports the performance of various methods on the Rendition and Sketch datasets. In this scenario, SSA significantly improved ROID and CMF performance, achieving state-of-the-art results. These results provide empirical evidence that SSA is not only effective in addressing data corruption but also robust in handling natural distribution shifts, reinforcing its reliability in diverse adaptation environments.

## B.3. Additional Experiments: Real-world Scenario

*Table 10.* Average word error rates (%) and standard deviation in the real-world speech recognition scenario on TED.

| Method | Source | Pseudo Label | TENT | CMF | SSA |
|---|---|---|---|---|---|
| Avg. WER | 12.4±0.00 | 12.1±0.03 | 11.9±0.02 | 11.8±0.05 | **11.5±0.01** |

Table 11 compares the performance of various TTA methods in a real-time speech recognition scenario on TEDLIUM3 (Hernandez et al., 2018), which consists of streamed talk recordings. The TEDLIUM3 test dataset includes speeches by 11 experts, each delivering talks on different topics. We used the speech version of D2V that was pre-trained in LibriSpeech (Panayotov et al., 2015). Following the CMF setup, we measured the average word error rate (WER) of the model in an environment where multiple speakers sequentially deliver speech. These results show that SSA achieved the best performance compared to competing TTA methods, demonstrating its effectiveness in adapting to speaker variations in real-time speech recognition.

## B.4. Additional Analysis: Various Model Architectures

*Table 11.* Average error rates (%) and standard deviation for various model architectures in covariate shift on ImageNet-C.

| Model | Source | ROID | CMF | SSA |
|---|---|---|---|---|
| ViT | 60.2 | 45.0±0.08 | 44.9±0.08 | **44.6±0.05** |
| Swin | 64.0 | 47.2±0.15 | 46.6±0.12 | **46.0±0.07** |
| D2V | 51.8 | 44.8±0.04 | 43.5±0.04 | **42.2±0.16** |

Table 11 compares the performance of TTA methods across different model architectures in the covariate shift scenario. The results show that SSA achieved state-of-the-art performance across all tested architectures, demonstrating its adaptability and effectiveness. Notably, SSA remained highly effective even for models with lower source model performance, such as Swin, where it achieved substantial improvements. This suggests that SSA is particularly beneficial where the base model struggles with adaptation.

## B.5. Additional Analysis: Validity of Steady-State Scale Factor

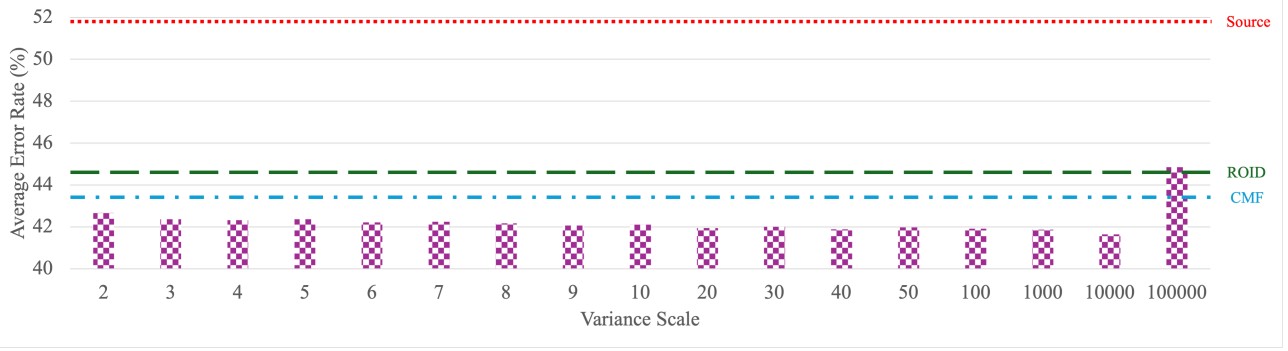

*Figure 5.* Average error rates (%) for various the steady-state scale factor.

The steady-state scale factor $\sigma_\lambda^2$ determines the baseline for covariance regulation in SSA. To analyze its impact, we varied this factor by multiplying $\sigma_\lambda^2$ by $\log(\text{Variance Scale})$, adjusting the scale accordingly. Figure 5 presents the average error rates in the covariate shift scenario for different variance scales. The results show that SSA maintained consistent performance for Variance Scale values ranging from 2 to 10,000, demonstrating its robustness in a wide range of covariance conditions. These results highlight the reliability and adaptability of SSA, confirming that its performance remains stable even under significant variations in the steady-state scale factor.

## C. Limitation and Future Works

One limitation of our current approach was the reliance on a linear Gaussian model to represent the weight distribution. Although this choice provides computational efficiency and analytical tractability, it can limit the flexibility of the model in capturing more complex weight behaviors. However, our empirical results demonstrated that SSA significantly improves performance in various scenarios, model architectures, and TTA methods with fixed hyperparameters. For future work, we aim to explore weight distributions that can model a broader range of weight dynamics. By extending beyond the linear Gaussian assumption, we seek to enhance the adaptability and robustness of our approach in more complex and rapidly evolving environments.

