# OpenReview forum: "Bayesian Weight Enhancement with Steady-State Adaptation for Test-time Adaptation in Dynamic Environments"
_ICML.cc/2025/Conference — ICML 2025 poster_

### Official Review · Reviewer_Tysa · 2025-03-03

**Overall Recommendation:** 3

**Summary:**

This paper proposes steady-state adaptation (SSA), a novel test-time adaptation (TTA) method that can be combined with existing ones.
SSA aims to reduce noise accumulation in gradients caused by the unsupervised nature of a TTA loss (e.g., entropy).
SSA models the distribution of the model weights and estimates it using the Bayesian weight enhancement.
Experimental results show that SSA improved classification accuracy under image corruption, label shifts, and domain shifts.

## update after rebuttal
I appreciate the author's further experiment. My concerns have been addressed. I have updated my score to 3.

**Claims And Evidence:**

Existing TTA methods accumulate noise in gradients due to the unsupervised nature of TTA losses, which results in the model weights being no longer reliable (weight degradation).
SSA robustly estimates the posterior weights by modeling their distribution by Gaussian, which improves the stability and efficiency of TTA.
Experimental results show that existing TTA methods degrade accuracy when adaptation is continually performed, while SSA retains high accuracy.

**Essential References Not Discussed:**

BACS[a] also models the posterior weight distribution of the model using SWAG[b].

[a] Zhou and Levine, Bayesian Adaptation for Covariate Shift, NeurIPS 2021.
[b] Izmailov et al., Averaging Weights Leads to Wider Optima and Better Generalization, UAI 2018.

**Experimental Designs Or Analyses:**

- In Tables 1 to 3, combining SSA with not only ROID and CMF but also the other methods would be interesting. Specifically, combining SSA with simple TTA methods like TENT would enhance SSA's generality and efficacy, which would also support the hypothesis.
- A comparison of each method's performance with and without SSA would provide valuable insights.
- The accuracy gains achieved by SSA (e.g., ROID vs. ROID+SSA, CMF vs. CMF+SSA) appear to be marginal.
- Experiments on other tasks, e.g., semantic segmentation, would strengthen the SSA's generality.

**Methods And Evaluation Criteria:**

Strength: Modeling and updating the posterior weights with the Bayesian weight enhancement is compelling and intriguing.

Weakness: The modeling is too simple. Does it sufficiently represent the weight distribution?
  - According to Eq. (6), $p({w}_{k+1}|{u})$ does not affected by ${u}$.
  - Is it sufficient to represent the covariance with a single scalar as $\Sigma_k=\sigma_k^2 \mathbf{I}$? Intuitively, each model weight has a different variance. For example, Modeling the covariance as diagonal, i.e., $\Sigma_k = \text{diag} (\sigma_{k,1}^2,\ldots, \sigma_{k,d}^2)$ would be interesting.

**Other Comments Or Suggestions:**

There are many typos, which are confusing:

- Eq. (1): $\hat{{w}\_o} \rightarrow \hat{{w}}\_0$
- Eq. (5): ${w}\_{+1} \rightarrow {w}\_{k+1}$
- Eq. (16): Is the $\int$ missing?
- The right column of L315, L658, etc.: $1.0^{-12} \rightarrow 10^{-12}$?
- Title: Stead -> Steady

**Other Strengths And Weaknesses:**

N/A

**Questions For Authors:**

N/A

**Relation To Broader Scientific Literature:**

The Bayesian weight enhancement can be integrated with various TTA methods.
It can raise the baseline of existing TTA literature.

**Theoretical Claims:**

I have checked the derivation of SSA using the SDE approximation of SGD.

Explaining how the proposed method addresses label shifts would be interesting.

---

> ### Author Rebuttal · Authors · 2025-03-29
>
> We sincerely appreciate your valuable review and insightful comments. We have carefully examined your feedback, fully understood the concerns and questions you raised, and have made every effort to reflect them thoroughly and faithfully in our manuscript.
>
> **Methods And Evaluation Criteria:**
>
> > According to Eq. (6), $p({w}_{k+1}|{u})$ does not affected by ${u}$.
>
> → In Eq. (6), the mean, represented by the source weight $w_0$, corresponds to the enhanced weight $u_0$ at time step 0, and thus is indirectly influenced by $u$.
>
> > Is it sufficient to represent the covariance with a single scalar as $\Sigma_k=\sigma_k^2 \mathbf{I}$? Intuitively, each model weight has a different variance.
>
> → In Appendix D, we explicitly mention exploring more complex and potentially better-suited weight distributions for the TTA process as future work, which includes considering the diagonal covariance structure suggested by the reviewer. However, the rationale behind retaining a scalar covariance in our current work is motivated by the considerable noise inherent to unsupervised learning within the TTA process. Under such noisy conditions, scalar covariance helps mitigate potential overfitting from covariance flexibility. The following results substantiate this claim:
>
> | Method | covariate shifts | label shifts ($\gamma=0.1$) | label shifts ($\gamma=0.0$) |
> | --- | --- | --- | --- |
> | SSA (scalar) | **42.2±0.16** | **13.1±0.29** | **35.9±0.04** |
> | SSA (diagonal) | 42.6±0.11 | 13.2±0.30 | 36.3±0.03 |
>
> **Theoretical Claims:**
>
> > Explaining how the proposed method addresses label shifts would be interesting.
> >
>
> → Under label shift scenarios, models often converge to trivial solutions, excessively increasing the probability assigned to specific classes. In such situations, gradients become excessively large, causing model weights to enter narrow optimal points (Section 3.3). SSA prevents the model from falling into narrow optimal points by decreasing the learning rate in response to gradients that significantly enlarge the variance.
>
> **Experimental Designs Or Analyses:**
>
> > Specifically, combining SSA with simple TTA methods like TENT would enhance SSA's generality and efficacy, which would also support the hypothesis.
> >
>
> → In Table 5 of Section 4, we have demonstrated the effectiveness and compatibility of combining SSA with TENT in the covariate-shift scenario. As shown in that table, even when combined with TENT, SSA yields a substantial performance improvement.
>
> → Additionally, beyond the covariate-shift scenario presented in Table 5, we measured the average error rates (%) for TENT combined with SSA in the label-shift scenarios (corresponding to Tables 2 and 3) as follows:
>
> | Method | covariate shifts | label shifts ($\gamma=0.1$) | label shifts ($\gamma=0.0$) |
> | --- | --- | --- | --- |
> | TENT | 53.3±0.22 | 55.4±1.58 | 56.2±0.98 |
> | TENT+SSA | **49.3±0.30** | **49.9±0.03** | **50.3±0.02** |
>
> **Essential References Not Discussed:**
>
> → In Appendix C (Bayesian Deep Learning), we have discussed SWAG [b], as you have suggested. SWAG has demonstrated robustness to out-of-distribution data, particularly when considering multiple models. Based on SWAG, BACS [a] obtains a posterior distribution over the data by marginalizing parameters through a prediction ensemble. However, such an approach is computationally inefficient because it requires making $N$ predictions for each inference, using $N$ models stored beforehand.
>
> → In contrast, SSA directly infers a time-varying posterior weight distribution in environments subject to gradient noise. It achieves this by performing Bayesian filtering through dynamics derived via an SDE approximation. Subsequently, SSA selects a single weight from the posterior distribution, thus requiring only one prediction per inference. Therefore, our proposed algorithm enables significantly more efficient inference than existing methods. Furthermore, SSA theoretically derives and accounts for the temporal evolution of the weight distribution (Section 3.3). We will include this discussion explicitly in Appendix C to further clarify and strengthen the contribution of our manuscript.
>
> **Other Comments Or Suggestions:**
>
> We sincerely appreciate your careful proofreading for typographical errors. Following your valuable suggestions, we have corrected the identified errors to enhance the readability of our manuscript. However, for the following cases, we either maintained our original notation or revised it differently from your suggestion for the reasons explained below:
>
> → In Eq. (16), we first computed the joint distribution $p(w_{k+1},u_{k+1}|w_{0:k})$ and then converted it into $p(u_{k+1}|w_{0:k},w_{k+1})=p(u_{k+1}|w_{0:k+1})$ with $Z_k=p(w_{k+1}|w_{0:k})$, which aligns with Bayes' rule. Hence, we decided to maintain our original notation.
>
> → Regarding $1.0^{-12}$, this notation was intended to represent $1^{-12}$. We have revised this to indicate $10^{-13}$ to prevent potential confusion.

---

> > ### Comment · Reviewer_Tysa · 2025-04-02
> >
> > I appreciate the author's response and additional experiments.
> >
> > > → In Table 5 of Section 4, we have demonstrated the effectiveness and compatibility of combining SSA with TENT in the covariate-shift scenario.
> >
> > I'm curious about the versatility of SSA.
> > Can SSA be combined with other methods, e.g., LAME, RoTTA, SAR, and EATA?
> > Adding +SSA score for each method to Tables 1-3 would be interesing.
> >
> >
> > > Regarding $1.0^{-12}$, this notation was intended to represent $1^{-12}$. We have revised this to indicate $10^{-13}$ to prevent potential confusion.
> >
> > $1^x=1$ regardless of $x$. Is the notation really correct?
> >
> >
> > -----
> > 6 Apr 2025
> >
> > I appreciate the author's further experiment. My concerns have been addressed. I have updated my score to 3.

---

> > > ### Author Response · Authors · 2025-04-03
> > >
> > > Thank you for your thoughtful and detailed review of our manuscript. We sincerely appreciate your insightful questions and would like to provide the following responses:
> > >
> > > - **Additional Experiments**
> > >
> > > → LAME is a parameter-free method, and RoTTA is a consistency training approach based on the student-teacher framework (see Appendix B.1). Therefore, they are not encompassed by the Bayesian weight enhancement framework that forms the basis of SSA. Excluding these two methods, the results of applying SSA to SAR and EATA are as follows:
> > >
> > > | Method | covariate shifts | label shifts ($\gamma=0.1$) | label shifts ($\gamma=0.0$) |
> > > | --- | --- | --- | --- |
> > > | SAR | 51.3±0.35 | 48.0±0.10 | 50.5±1.38 |
> > > | SAR+SSA | **49.0±0.09** | **47.3±0.21** | **46.6±0.08** |
> > > | EATA | 50.2±0.06 | 43.3±0.03 | 47.0±0.11 |
> > > | EATA+SSA | **47.6±0.06** | **41.3±0.02** | **45.2±0.01** |
> > >
> > > → As your insightful anticipation suggested, SSA demonstrates strong compatibility and generality when combined with a variety of existing methods in improving performance across both SAR and EATA, under various types of distribution shifts .
> > >
> > > - **Notation**
> > >
> > > → Thank you for pointing out. Following your suggestion, we have revised the notation to $10^{-12}$ accordingly.
> > >
> > > Your detailed review has significantly contributed to improving the rigor and clarity of our manuscript. Once again, we are deeply grateful for your careful evaluation and constructive feedback.

---

### Official Review · Reviewer_JHQ8 · 2025-03-11

**Overall Recommendation:** 3

**Summary:**

This manuscript proposes a novel Bayesian-based framework to enhance existing weight-based TTA methods. They investigate the distribution shifts issue and reflect the reason behind gradient noise. A tailored steady-state adaptation algorithm shows SOTA performance on several benchmarks.

**Claims And Evidence:**

The claims made in the submission are well-supported by a robust analytical framework and experimental results, which collectively validate the effectiveness of the work.

**Essential References Not Discussed:**

None

**Experimental Designs Or Analyses:**

The authors have conducted several experiments that demonstrate the ability of the proposed SSA algorithm to enhance the generalization capabilities of recent ROID and CMF methods. The experimental design appears to be comprehensive, with thorough consideration of various submodules and hyperparameters.

**Methods And Evaluation Criteria:**

1. The proposed framework employs the SDE approximation to ensure steady-state covariance, which can align with the discrete TTA process.
2. The authors design a dynamic algorithm for the step size calculation, which can balance covariance based on the posterior weight distribution.

**Other Comments Or Suggestions:**

None.

**Other Strengths And Weaknesses:**

+ The manuscript is well-organized with clear illustrations.

**Questions For Authors:**

An additional issue is that, based on the experimental data, the performance improvement of SSA on ROID appears to be less significant than that of CMF. Have the authors analyzed the reasons for this difference?

**Relation To Broader Scientific Literature:**

The study of online TTA presented in this paper is of significant relevance to both computer vision and protein prediction fields. However, the paper's exclusive focus on weight-based methods limits the scope of its contributions and may overlook other potential approaches.

**Theoretical Claims:**

Yes.

---

> ### Author Rebuttal · Authors · 2025-03-29
>
> We deeply appreciate your valuable review and insightful comments. We have carefully considered your feedback, fully understood the concerns and questions raised, and endeavored to address them thoroughly and sincerely.
>
> **Relation To Broader Scientific Literature:**
>
> > However, the paper's exclusive focus on weight-based methods limits the scope of its contributions and may overlook other potential approaches.
> >
>
> → As you rightly pointed out, our study primarily focuses on the mechanisms underlying weight-based methods. Nonetheless, our contribution goes beyond merely analyzing weight-based techniques; we uncover and advance the underlying probabilistic framework. By doing so, we effectively mitigate instability issues commonly encountered in TTA methods, especially those caused by increasing learning rates (Section 5). Furthermore, as demonstrated in Table 5, the SSA method has shown notable generality by significantly improving performance when combined with fundamental TTA methods such as TENT.
>
> **Questions For Authors**
>
> > An additional issue is that, based on the experimental data, the performance improvement of SSA on ROID appears to be less significant than that of CMF. Have the authors analyzed the reasons for this difference?
> >
>
> → The issue raised by the reviewer can be explained through the discussion presented in Appendix A.1. As mentioned in that section, CMF provides a distribution for the hidden source weights that evolves in real-time based on observed weights. This approach contrasts with the typical scenario in Eq. (16), where the mean of the likelihood $p(w_{k+1}|u_{k+1})$ is fixed as the source weights (i.e., combining ROID with SSA). In other words, CMF's modeling of the hidden source weights facilitates a more sophisticated likelihood estimation.
>
> → This advantage explains why the combination of CMF and SSA achieves superior performance. We will incorporate this explanation into Section 4.2 of the manuscript to clarify further the superior performance of the CMF and SSA combination, thereby enhancing the robustness of our paper.

---

### Official Review · Reviewer_a4NL · 2025-03-15

**Overall Recommendation:** 3

**Summary:**

This paper proposes using a stochastic differential equation (SDE) to handle temporal distribution shifts in test-time adaptation scenarios. The SDE is applied to handle the temporal dynamics of stochastic gradient descent, balancing the current updates of the model weight with that of the pre-trained model. The experiment results give the performance gains of the proposed method over two datasets.

**Claims And Evidence:**

One of the key claims is a "Bayesian weight enhancement framework that unifies and generalizes existing weight-based TTA methods." However, the approach appears to primarily apply an SDE to SGD dynamics, which was a major contribution of prior work (Li et al., 2019). This work extends it by introducing Bayesian filtering to estimate the posterior of weight distributions.

**Essential References Not Discussed:**

The key contribution lies in the application of SDE or a dynamic system to address online distribution shifts in TTA. However, this general idea has been explored before, albeit with different definitions of SDE or dynamic systems. For example:

- Huang et al. (2022): Extrapolative continuous-time Bayesian neural network for fast training-free test-time adaptation (NeurIPS 2022).
- Schirmer et al. (2024): Test-time adaptation with state-space models (ICML 2024 Workshop on Structured Probabilistic Inference & Generative Modeling).
While this work builds on these foundations, the novelty may be limited given prior investigations in similar directions. But I do appreciate the practical contribution of this work.

**Experimental Designs Or Analyses:**

- The method achieves state-of-the-art (SoTA) results on two datasets, ImageNet-C and D109. However, a more in-depth analysis is lacking, particularly regarding the computational cost introduced by the new method or analysis that indicates noise covariance has been effectively handled.

- The ablation study is insufficient. For example, is Bayesian filtering truly necessary (empirically and theoretically), or would directly applying the SDE from Li et al. (2019) suffice?

**Methods And Evaluation Criteria:**

The proposed methods make sense, and the datasets for evaluation seem reasonable.

**Other Comments Or Suggestions:**

N.A.

**Other Strengths And Weaknesses:**

See above.

**Questions For Authors:**

See above.

**Relation To Broader Scientific Literature:**

In the broader scientific context, this work builds upon SDE+SGD (Li et al., 2019) by incorporating Bayesian filtering to handle distribution shifts in TTA. Since both components are well-established with existing solutions, the theoretical contribution remains unclear. While combining these two techniques is reasonable, the overall contribution does not appear particularly significant.

**Theoretical Claims:**

N.A.

---

> ### Author Rebuttal · Authors · 2025-03-29
>
> We sincerely appreciate your valuable review and insightful comments. We have carefully examined your feedback, fully understood the concerns and questions raised, and have earnestly endeavored to address them.
>
> **Claims And Evidence:**
>
> > However, the approach appears to primarily apply an SDE to SGD dynamics, which was a major contribution of prior work (Li et al., 2019).
>
> **Experimental Designs Or Analyses:**
>
> > The ablation study is insufficient. For example, is Bayesian filtering truly necessary (empirically and theoretically), or would directly applying the SDE from Li et al. (2019) suffice?
> >
>
> → Below, we address your comments across Claims And Evidence, Experimental Designs Or Analyses. We understood your comments to revolve around questioning whether modeling the TTA process solely via SDE approximating SGD is sufficient. Relying exclusively on the SDE approximation is inadequate for capturing weight evolution based on discrete observations, and it prevents the use of empirically validated weight-based TTA methods. The detailed reasoning is as follows:
>
> - **The SDE approximation** describes how weights evolve without data during the TTA process.
>     - Additionally, while the SDE approximation provides dynamics in continuous time, actual observations occur only at discrete time steps $k$, necessitating the discretized transition distribution we derived in Eq. (12).
> - **Bayesian filtering** explains how the TTA process evolves given observations (i.e., weights).
>     - The TTA process deals explicitly with noisy observations (Section 3.1). To model these noisy observations effectively, it is essential to interpret weight-based TTA methods probabilistically through Bayesian weight enhancement, as discussed in Section 3.2. This necessity results in the likelihood $p(w_{k+1}|u_{k+1})$ in Eq. (16), implicitly incorporating noise.
>     - Combined with the transition distribution, this likelihood integrates into Bayesian filtering in Eq. (13) and Eq. (16), thus yielding the posterior weight distribution incorporating past observations.
>
> → These points underscore the significance of integrating SDE approximation with Bayesian filtering and highlight our contributions.
>
> **Experimental Designs Or Analyses**
>
> > However, a more in-depth analysis is lacking, particularly regarding the computational cost introduced by the new method or analysis that indicates noise covariance has been effectively handled.
> >
>
> → As shown in Algorithm 1, the SSA method requires only simple arithmetic operations among weights, resulting in minimal additional computational overhead. In Table 5, we addressed your concern regarding computational cost by analyzing GPU wall time and relative execution time before and after applying SSA. Our analysis reveals that SSA incurs only an additional relative computational cost ranging from 1% to 5%, depending on the specific approach.
>
> → The SSA method incorporates a mechanism to drive the noise covariance towards a steady-state over time. As demonstrated in Figure 4, we measured the posterior covariance over time for TENT and CMF methods without SSA applied. Our results show that existing methods exhibit significant temporal fluctuations in covariance. In contrast, such fluctuations decrease significantly when SSA is used, and the covariance converges towards a particular steady-state value.
>
> **Essential References Not Discussed**
>
> > The key contribution lies in the application of SDE or a dynamic system to address online distribution shifts in TTA. However, this general idea has been explored before, albeit with different definitions of SDE or dynamic systems. For example:
>
> → In Appendix C (Bayesian Filtering), we have cited the reference you recommended [Huang et al. (2022)]. This reference employs a particle filtering approach, which requires offline training using source and target data and a sampling process to learn the distribution parameters of weights and their importance. In contrast, SSA uses only online learning and requires only one sampling, resulting in significantly higher computational efficiency (Section 3.3).
>
> → Another reference you suggested [Schirmer et al. (2024)] utilizes a state-space model to capture the distribution of representation and weight changes. Consequently, their method leads to a complex design for the posterior of the weight distribution, resulting in additional computational overhead. On the other hand, SSA directly infers the posterior of the weight distribution using a transition model derived by introducing an SDE approximation. This approach allows SSA to perform inference through simple arithmetic operations, achieving high computational efficiency (Table 5). We will include this discussion in Appendix C (Bayesian Deep Learning) to clarify the contributions of SSA further.

---

> > ### Comment · Reviewer_a4NL · 2025-04-04
> >
> > Most of my concerns have been resolved. However, in this work, the SDE approximation seems to function as a prior (am I understanding right ?), not incorporating data or observations during the TTA process. It does, however, include some methods for using observations to parameterize the variables in the SDE. Additionally, discretization can be achieved by solving the equations at discrete time intervals. Nonetheless, it is still necessary to conduct an ablation study to clarify the contribution of each component.

---

> > > ### Author Response · Authors · 2025-04-06
> > >
> > > → Thank you for your constructive comments. We can interpret the transition distribution derived from the SDE as serving as a prior over the weights over time. Specifically, the transition distribution enables Bayesian filtering to infer the posterior weight distribution at the current time step by integrating past weight information. This posterior then acts as the prior distribution at the next time step (see Section 3.3, “Online Posterior Weight Distribution Inference”). We derived the balancing covariance using the posterior distribution and steady-state condition (Section 3.3, “Balancing Covariance”).
> > >
> > > → In light of this background, we have conducted the additional ablation study you suggested on the balancing covariance and the transition distribution. The results are summarized as follows:
> > >
> > > | Method | covariate shifts | label shifts ($\gamma=0.1$) | label shifts ($\gamma=0.0$) |
> > > | --- | --- | --- | --- |
> > > | SSA | 42.2±0.16 | 35.9±0.04 | 13.1±0.29 |
> > > | No Balancing Covariance | 43.0±0.20 | 37.9±0.10 | 14.0±0.31 |
> > > | No Transition Distribution | 43.5±0.04 | 38.2±0.05 | 14.4±0.24 |
> > >
> > > → Thanks to your insightful suggestion, the individual contributions of each component in our method have become more apparent. Once again, we sincerely appreciate your thoughtful feedback.

---

### Official Review · Reviewer_HHYp · 2025-03-15

**Overall Recommendation:** 4

**Summary:**

Test-time adaptation assumes that only the inputs of the test dataset are given for adaptation, where the model parameters are updated using an unsupervised loss without labels. Consequently, the model parameters are inevitably updated by a noisy gradient, which differs from the gradient obtained using true labels. Thus, this work considers the weight parameters as random variables and applies the Kalman filtering approach to update the mean and covariance of the random weight parameters. Unlike previous Bayesian approaches, this work presents how to adapt the learning rate/step size of the mean using the covariance so that the weight parameters are updated in a way that ensures the covariance of the noisy gradient remains stable, meaning it does not change significantly compared to its previous update. Empirically, the effectiveness of the proposed method is demonstrated on ImageNet using various scenarios of covariate shift and label shift, as well as in terms of learning rate robustness.

**Claims And Evidence:**

The claim is well supported by empirical results.

**Essential References Not Discussed:**

N/A

**Ethical Review Flag:**

Flag this paper for an ethics review.

**Experimental Designs Or Analyses:**

The experimental design and analysis are well conducted.

**Methods And Evaluation Criteria:**

The proposed method seems to make sene.

**Other Comments Or Suggestions:**

N/A

**Other Strengths And Weaknesses:**

### Strengths

*  This work recognizes an important problem in previous test-time adaptation algorithms, where parameter updates using noisy gradients could result in performance degradation of the adapted model due to an improperly tuned learning rate.

* This work presents a clever way to address the learning rate issue by considering that the filtered covariance of the noisy gradient remains stable, and then determining the adaptive step size depending on the filtered covariance.

### Weaknesses

* In my view, this work does not seem to have any clear weaknesses.

**Questions For Authors:**

I do not have any further question on this.

**Relation To Broader Scientific Literature:**

This work presents a clever approach to updating model parameters in test-time adaptation while considering a realistic adaptation setting. In this context, the proposed method has the potential to enable DNN models to adapt effectively and continually to new environments.

**Theoretical Claims:**

Although this work does not have the theoretical claim, the claim of the proposed scala parameter in Eq. (19) seems valid.

---

> ### Author Rebuttal · Authors · 2025-03-29
>
> We deeply appreciate your valuable review and insightful comments. In line with your suggestions, we are extending our research to address a general problems of DNNs. Once again, thank you very much for your thoughtful feedback.

---

### Decision · Program_Chairs · 2025-05-01

**Decision:**

Accept (poster)

**Comment:**

The work addresses the weight degradation problem in unsupervised Test-time Adaptation by proposing a Bayesian approach with steady-state adaptation which ensures the covariance of the noisy gradient remains stable. Reviewers agree that the idea is interesting and the effectiveness of the method is well supported by real data experiments under various scenarios of distribution shift. Authors are encouraged to conduct ablation study to clarify the contribution of each component of their proposed method.